# ECG Electrode Localization: 3D DS Camera System for Use in Diverse Clinical Environments

**DOI:** 10.3390/s23125552

**Published:** 2023-06-13

**Authors:** Jennifer Bayer, Christoph Hintermüller, Hermann Blessberger, Clemens Steinwender

**Affiliations:** 1Institute for Biomedical Mechatronics, Johannes Kepler University, 4040 Linz, Austria; 2Department of Cardiology, Kepler University Hospital, 4020 Linz, Austria; 3Medical Faculty, Johannes Kepler University, 4020 Linz, Austria

**Keywords:** electrode localization, 3D camera, real-time 3D recording, exposure control, white balancing, system calibration, image processing, surface alignment

## Abstract

Models of the human body representing digital twins of patients have attracted increasing interest in clinical research for the delivery of personalized diagnoses and treatments to patients. For example, noninvasive cardiac imaging models are used to localize the origin of cardiac arrhythmias and myocardial infarctions. The precise knowledge of a few hundred electrocardiogram (ECG) electrode positions is essential for their diagnostic value. Smaller positional errors are obtained when extracting the sensor positions, along with the anatomical information, for example, from X-ray Computed Tomography (CT) slices. Alternatively, the amount of ionizing radiation the patient is exposed to can be reduced by manually pointing a magnetic digitizer probe one by one to each sensor. An experienced user requires at least 15 min. to perform a precise measurement. Therefore, a 3D depth-sensing camera system was developed that can be operated under adverse lighting conditions and limited space, as encountered in clinical settings. The camera was used to record the positions of 67 electrodes attached to a patient’s chest. These deviate, on average, by 2.0 mm ±1.5 mm from manually placed markers on the individual 3D views. This demonstrates that the system provides reasonable positional precision even when operated within clinical environments.

## 1. Introduction

Models of the human body have gained increasing interest in clinical research and are essential for delivering personalized diagnoses and treatments to patients. They can be used to build a digital twin of a patient’s body that can be used for planning curative interventions, predicting the outcomes of intended treatments, or the likelihood of relapses and complications. For most of these models, apart from the knowledge of the patient’s exact anatomy, information about physiological processes is required such as the impedance of fibrous tissue forming an infarction scar, which largely differs from the impedance of intact myocardium.

Electrical impedance tomography (EIT) and electrical capacitance tomography (ECT) are used to measure tissue parameters such as impedance or capacitance [1,2,3]. The origin of cardiac arrhythmias or myocardial infarctions can be identified by integrating ECG recordings [4,5,6], and the functions within the human brain [7,8] can be visualized using models, including EEG recordings. All these methods require the positions of between 12 and a few hundred sensors to be exactly known. The larger the positional error, the lower the diagnostic value of the results generated by the model. Consequently, they are less suitable for treatment planning, guidance, outcome stratification, or prevention of complications and relapses.

A commonly used approach is to extract the sensor positions, along with the anatomical details, from Magnet Resonance Image (MRI) stacks or X-ray Computed Tomography (CT) slices. Both approaches require special markers to be attached to the sensors, which are visible in MRI [9,10] or CT scans [11]. Identifying the sensor positions from MRI or CT scans yields the smallest positional errors compared to the true sensor position. However, this approach significantly hinders the clinical uptake and widespread use of electrical impedance tomography (EIT), electrical capacitance tomography (ECT), noninvasive imaging of cardiac electrophysiology (NICE), and other model-based approaches. Patients either have to be exposed to large amounts of ionizing radiation when using CT scans, limiting the use of the aforementioned methods to three applications per year. Although MRI is not bound by this limitation, it is only covered by insurance companies if it is required for obtaining a proper diagnosis and evaluating outcomes.

Given these limitations, alternative approaches that decouple the generation of the underlying anatomical models from the localization of the sensors have been tested [12,13,14]. Alternatives such as magnetic digitizer systems, e.g., the Polhemus Fastrak [12], tracked gaming controllers [13], or motion capture systems, have been used to identify the positions of electrodes relative to the patient’s body. The use of photogrammetry, visual odometry, and stereoscopic approaches was already considered more than 15 years years ago [15,16]. The Microsoft Kinect 3D depth-sensing camera (3D DS) was one of the first compact and affordable devices. Nowadays, modern coded light and stereo vision-based models are portable and lightweight enough to be easily attached to or even integrated within a standard tablet computer.

In the past few decades, 3D DS cameras have mainly been used in EEG-based studies to locate EEG sensors on the patient’s skull [12,14,17]. All of them use the recorded EEG signals to localize brain activity or identify the focus of a seizure within the cortex. In contrast, very few studies report the use of 3D DS cameras to locate ECG sensors on the chest or even the whole torso [18,19,20]. One reason for this may be that the skull is a rigid structure that does not change its shape when the subject moves during the recording. In contrast, when recording the sensor position on the torso, the patient needs to maintain a specific posture. The instructions provided to the patient on how to achieve and maintain this posture are integral to the entire recording procedure.

In the present work, the positions of 64 ECG electrodes mounted on the torso are recorded using 3D DS camera readings only. Section 2 encompasses descriptions of the overall structure of the developed 3D DS camera-based system, method, and algorithm used for the real-time recording of the individual 3D views of the torso (Section 2.2); the postprocessing steps necessary for extracting the electrode positions (Section 2.3); and the recording protocol used and the instructions provided to each subject participating in the clinical testing (Section 2.4). In Section 3, the results obtained from the five subjects are presented, and in Section 4, these results are discussed.

## 2. Materials and Methods

The 3D depth-sensing (3D DS) camera-based measurement of electrode positions can be divided into four main steps: (i) selecting the appropriate 3D DS camera, (ii) defining an appropriate measurement protocol, (iii) recording the 3D surfaces in real time, and (iv) extracting the electrode center points.

The most important component for recording the electrode positions is the 3D camera. It can be characterized by various parameters such as the closest distance $d_{near}$ and the vertical $\psi_{V}$ and horizontal $\psi_{H}$ fields of view (FOV). These parameters define the volume in front of the camera in which objects must be placed to be accurately captured by the depth sensor. Based on these considerations, the Intel Realsense SR300 3D DS camera [21] was selected. Descriptions of the exact selection criteria that led to this decision can be found in Section 2.1.

The human torso represents a flexible object that offers several degrees of freedom for movement and deformation in contrast to the rather rigid skull. The position of each ECG electrode perceived by the 3D DS camera and its relative position to the other electrodes is directly affected by the movements of the patient’s body. Therefore, it is essential to define an appropriate recording protocol before the first 3D data set is recorded. As large displacements may prevent the successful extraction of the electrode positions, the patient is required to actively maintain the same posture throughout the recording procedure. Details on how this active engagement of the patient can be achieved are described in Section 2.4. For the remaining steps, (iii) real-time recording and (iv) offline processing, Figure 1 provides an overview of the necessary sub-steps and their interdependence:

A 3D DS camera combines a depth sensor and an RGB color sensor in a single device. These two sensors simultaneously record an RGB color image $\Xi_{rgb}$ and a 16-bit depth image $D_{16}$. The latter image encodes the distance *d* between the camera and the objects located in front of the camera.

The developed real-time recording system is intended for use in diverse clinical settings such as examination rooms in outpatient clinics or local cardiology practitioner clinics. The lighting conditions encountered depend on the pointing direction of the camera and the number of light sources, as well as their brightness and color hue. In order to properly handle these conditions, the white-balancing settings, exposure time τ, and overall gain $\gamma_{ex}$ of the color sensor are continuously adjusted in real time. Automatic white balancing (AWB), which is described in detail in Section 2.2.1, uses $\Xi_{rgb}$ to estimate the color temperature $K_{W}$ of the dominant light source.

At the same time, a binary mask $M_{D}$ is generated from the depth image $D_{16}$. This $M_{D}$ splits $D_{16}$ into foreground pixels representing the torso surface and objects in the background (Section 2.2.3). $M_{D}$ is used to generate a 3D mesh *S* of the imaged torso surface (Section 2.2.4) and tune the exposure time τ and global gain setting $\Gamma_{ex}$ of the color sensor. This is achieved by combining $M_{D}$ with the brightness information *I* of the color image $\Xi_{rgb}$ obtained during the AWB step (Section 2.2.2). The mask $M_{D}$ is also used to outline the patient’s contours on the real-time preview screen, along with various system parameters.

When the trigger is pressed, the triangulation component (Figure 1) generates a 3D surface mesh *S*, which is stored along with the corresponding texture information $\Xi_{uvs}$ of the torso created from the RGB color image Ξ.

In the offline processing step, a pairwise iterative closest-point (ICP) algorithm is used to align the recorded surfaces *S* with each other. The resulting transformation matrices ***ℜ*** are used to extract the 3D positions from the color-corrected texture images $\Xi_{uvs}$, which have been stored alongside each *S* (Section 2.3.2). In order to facilitate the steps necessary to identify the color markers attached to the electrodes, an additional color-correction step, which is described in Section 2.3.1, is conducted. The aim of this step is to ensure that the patient’s skin color and marker colors are accurately represented across all the recorded texture images $\Xi_{uvs}$. To achieve this, the $\Xi_{uvs}$ are split into a chromacity image $\chi_{rg}$ and the corresponding intensity image *I*. Both are used to identify the red and blue pixels and related 3D points corresponding to each electrode marker. Details on how this is achieved can be found in Section 2.3.3.

The centers of these markers are coaligned with the centers of the electrode clips and patches. Their positions on the surface are computed by fitting a planar model (Section 2.3.4) to the extracted red and blue points. In the final labeling step (Section 2.3.5), the electrode positions are assigned to the corresponding ECG signals recorded from the patient’s torso.

The colors of the markers vary depending on the position and orientation of the electrode clip relative to the torso and 3D DS camera. Therefore, a dedicated calibration procedure is utilized, which is outlined in Section 2.3.6, to determine the ranges of the red and blue color values that represent the electrode markers.

### 2.1. Selecting the Camera

The selected Intel Realsense SR300 3D DS camera [21] is used in narrow or crowded places such as examination rooms in outpatient clinics and cardiology practitioner clinics. In these places, the patient is typically seated on an examination bed or chair placed close to the wall. Consequently, the closest distance $d_{near}$ relative to the depth sensor at which objects may be placed has to be shorter than the shortest horizontal distance $d_{H,min}$ of the patient’s torso to any surrounding obstacles such as walls or furniture. The horizontal $\psi_{H}$ and vertical $\psi_{V}$ FOVs determine how tall or wide the closest object can be to be fully captured in its height and width. The minimum required values for $\psi_{H}$ and $\psi_{V}$ can be approximated based on the patient’s approximated $h_{torso}$ and the 3D DS camera’s $d_{near}$ using the following relationships:(1)$$\begin{array}{c} d_{near}<d_{H,min}\\ \psi_{min}=2\arctan\left(\frac{h_{torso}}{d_{near}+d_{H,min}}\right)\\ \max\left(\psi_{H},\psi_{V}\right)>=\psi_{min} \end{array}$$

According to the datasheet [21] the depth sensor can capture objects located at distances between 20 cm and 150 cm from the camera. This range is more than sufficient to record the surface of the torso. The depth information of each object is captured using an infrared sensor in combination with a near-infrared projector [21,22].

The depth images $D_{16}$ are recorded in 4:3 image format, covering a horizontal FOV of 69 degrees and a vertical FOV of 54 degrees at a depth resolution of less than 1 mm. The color sensor of the camera generates the RGB images $\Xi_{rgb}$ in 16:9 format. Its horizontal FOV of 68 degrees is sufficiently well-paired with the horizontal FOV of the depth sensor. With a $\psi_{V}$ of 41 degrees, it covers only 2/3 of the depth sensor in height. This results in a lack of color information for the pixels close to the top and bottom edges of the depth image $D_{16}$, which was considered when outlining the measurement protocol in Section 2.4.

### 2.2. Real-Time Recording

#### 2.2.1. Automatic White Balancing

The color sensor used by the Intel Realsense 3D DS camera offers the possibility to manually tune the color gains $\Gamma_{\hat{R}}$, $\Gamma_{\hat{G}}$, $\Gamma_{\hat{B}}$ indirectly by adjusting their color temperature parameter $K_{W}$. This was used to implement a custom AWB component (Figure 1), along with the algorithm proposed in [23], which can handle these varying conditions. After applying a lookup table $\hat{v}=\left\lfloor v\right\rfloor$ based on linearization (gamma decompression) and normalization to the interval [0,1] of the red $\hat{R}=\left\lfloor R\right\rfloor$, green $\hat{G}=\left\lfloor G\right\rfloor$, and blue $\hat{B}=\left\lfloor B\right\rfloor$ color channels, the resulting linear RGB image Ξ is converted into an RGB chromacity $\chi_{rg}$ image and a linear grayscale image *I* that encodes the brightness of each pixel.
(2)$$\begin{array}{c} I_{i}={\hat{R}}_{i}+{\hat{G}}_{i}+{\hat{B}}_{i}\\ r_{i}={\hat{R}}_{i}/I_{i},\;g_{i}={\hat{G}}_{i}/I_{i},\;b_{i}={\hat{G}}_{i}/I_{i} \end{array}$$

From $\chi_{rg}$, all pixels $\left(r_{i},\;g_{i},\;b_{i}\right)$ are selected that encode shades of gray. The red *r*, green *g*, and blue *b* chromacity values of these pixels are located within a small area around the neutral color gamut point, which has a color temperature of 5500 K, as shown in Figure 2. The basic assumption is that these pixels most likely correspond to object surfaces of a neutral gray color. Consequently, a reddish-colored taint in these pixels must be caused by a low *K* value of the predominant illumination, and a bluish cast most likely results from a light source with a large *K*. Overexposed pixels are excluded, as their color most likely results from the saturation of at least one of the three color channels and thus does not properly represent the skin color of the patient or the color of the illuminant. Likewise, underexposed pixels are not considered, as their color is most likely caused by camera noise rather than the light reflected by the imaged object.

For adjusting the color temperature setting $K_{W}$ of the 3D DS camera, only pixels $\left(r_{i,W},\;g_{i,W},\;b_{i,W},I_{i,W}\right)$ that are located within a small area surrounding the neutral color gamut point are selected, which is, according to Cohen [23], defined by the chromacity values $\bar{r}=0.363$, $\bar{g}=0.338$, and $\bar{b}=0.299$. This area encloses all pixels that are located within the following two ellipses centered at the color gamut point: (3)$$\begin{array}{c} \frac{{\left(r_{i,W}-\bar{r}\right)}^{2}}{\sigma_{r}^{2}}+\frac{{\left(g_{i,W}-\bar{g}\right)}^{2}}{\sigma_{g}^{2}}<=1\;\mathrm{and}\;\frac{{\left(b_{i,W}-\bar{b}\right)}^{2}}{\sigma_{b}^{2}}+\frac{{\left(g_{i,W}-\bar{g}\right)}^{2}}{\sigma_{g}^{2}}<=1 \end{array}$$(4)$$\begin{array}{c} 0.00155I_{max}<I_{i}<0.955I_{max} \end{array}$$

Their primary and secondary axes are defined by the standard deviations for the red $\sigma_{r}=0.0723$, green $\sigma_{g}=0.0097$, and blue $\sigma_{b}=0.0749$ chromacity values with respect to the neutral color gamut point, which was determined in [23]. The maximum intensity encountered is maxI=3.

The lower $I_{min}=0.02$ and upper $I_{max}=0.98$ exposure limits, as defined for each channel in [23], are linearized to $I_{min}=0.02/12.92$ and $I_{max}={\left(\left(0.98+0.055\right)/1.055\right)}^{2.4}$ before applying them to the overall linear intensity values *I*.

To match $K_{W}$ with the color temperature *K* of the light source, the overall color gain $\Gamma_{K}$ of the camera is estimated. The following model is used to simulate how the camera adjusts the gain $\Gamma_{\hat{R}}$ of its red and blue $\Gamma_{\hat{B}}$ channels when $K_{W}$ is updated.
(5)$$\Gamma_{R}=\Gamma_{K},\;\Gamma_{\hat{B}}=\frac{1}{\Gamma_{K}},\;\Gamma_{K}=\frac{2(K_{W}-K_{W,min})}{K_{W,max}-K_{W,min}},\;K_{W}=K_{W,n+1}=\gamma K_{W,n}$$

Neither the lower and upper limits for $\Gamma_{\hat{R}}$ and $\Gamma_{\hat{B}}$, nor the color temperature corresponding to equal gain values $\Gamma_{\hat{R}}=\Gamma_{\hat{B}}=1$, are documented for up-to-date 3D DS cameras. It is assumed that $\Gamma_{\hat{R}}=\Gamma_{\hat{B}}=1$ corresponds to the center color temperature ${\bar{K}}_{W}=\left(K_{W,min}+K_{W,max}\right)/2$ between the minimum $K_{W,min}$ and maximum $K_{W,max}$ values of the color sensor. On startup, $K_{W}$ is initialized to $K_{W,0}={\bar{K}}_{W}$. For the recorded color images $\Xi_{n+1}$, the corresponding $K_{W,n+1}=\gamma K_{W,n}$ is estimated from the previous value of $K_{W}=K_{W,n}$ and a scaler γ reflecting the relative change of $K_{W}$ between two consecutive $\Xi_{n}$. The color sensor of the used camera has a rolling shutter. Therefore, color images are only considered for estimating the scaler γ and $K_{W}$ after the next exposure time interval has elapsed.

The goal is to minimize the distance between the average ${\bar{r}}_{W}$, green ${\bar{g}}_{W}$, and blue ${\bar{b}}_{W}$ chromacities of the selected pixels and the $\bar{r}$, $\bar{g}$, $\bar{b}$ of the color gamut point that corresponds to a color temperature of K=5500Kelvin. To achieve this, ${\bar{r}}_{W}$, ${\bar{g}}_{W}$, and ${\bar{b}}_{W}$ are multiplied by the unknown intensity $\dot{y}={\bar{R}}_{W}+{\bar{G}}_{W}+{\bar{B}}_{W}$ to obtain the corresponding mean red ${\bar{R}}_{W}$, green ${\bar{G}}_{W}$, and blue ${\bar{B}}_{W}$ color values. These values are scaled by γ using (Equation 5). After scaling, the updated ${\bar{r}}_{W}$, ${\bar{g}}_{W}$, and ${\bar{b}}_{W}$ are computed using (Equation 2).
(6)$$\bar{r}=\frac{{\bar{r}}_{W}\Gamma_{K}\dot{I}}{{\bar{r}}_{W}\Gamma_{K}\dot{I}+{\bar{g}}_{W}\dot{I}+\frac{{\bar{b}}_{W}\dot{I}}{\Gamma_{K}}},\;\bar{g}=\frac{{\bar{g}}_{W}\dot{I}}{{\bar{r}}_{W}\Gamma_{K}\dot{I}+{\bar{g}}_{W}\dot{I}+\frac{{\bar{b}}_{W}\dot{I}}{\Gamma_{K}}},\;\bar{b}=\frac{\frac{{\bar{b}}_{W}\dot{I}}{\Gamma_{K}}}{{\bar{r}}_{W}\Gamma_{K}\dot{I}+{\bar{g}}_{W}\dot{I}+\frac{{\bar{b}}_{W}\dot{I}}{\Gamma_{K}}}$$

It is obvious that the unknown intensity $\dot{I}$ does not have any impact on the result. It can be omitted from (Equation 6) and γ. Consequently, $\Gamma_{K}$ can be computed from $\bar{r}$, $\bar{g}$, $\bar{b}$, ${\bar{r}}_{W}$, ${\bar{g}}_{W}$, and ${\bar{b}}_{W}$ directly.

In Figure 2, it can be observed that the curve along which the color gamut point moves can be approximated for color temperatures K≤5000K by the line connecting the red corner (r=1,g=0,b=0) and the midpoint (r=0,g=0.5,b=0.5) between the blue (r=0,g=0,b=1) and green corners (r=0,g=1,b=0) of the chromacity space. For color temperatures K>5000K, the curve can be approximated by the line connecting the blue corner (r=0,g=0,b=1) with the midpoint (r=0.5,g=0.5,b=0) between the red (r=1,g=0,b=0) and green (r=0,g=1,b=0) corners, respectively. The two midpoints (r=0.5,g=0.5,b=0) and (r=0,g=0.5,b=0.5) correspond to the yellow $\bar{y}=\left(\bar{r}+\bar{g}\right)/2$ and cyan $\bar{c}=\left(\bar{g}+\bar{b}\right)/2$ chromacities, respectively. Based on the ratio $\bar{y}/\bar{b}$, the average chromacity value ${\bar{g}}_{W}\gamma$ of the green channel scaled by γ can be expressed. The resulting expression is inserted into the quadratic Equation (Equation 8) obtained from the ratio $\bar{r}/\bar{c}$: (7)$$\begin{array}{c} \frac{\bar{r}}{\bar{c}}=\frac{2\bar{r}}{\bar{g}+\bar{b}}=\frac{2{\bar{r}}_{W}\gamma }{{\bar{g}}_{W}+\frac{{\bar{b}}_{W}}{\gamma }},\;\frac{\bar{y}}{\bar{b}}=\frac{\bar{r}+\bar{g}}{2\bar{b}}=\frac{{\bar{r}}_{W}\gamma +{\bar{g}}_{W}}{2\frac{{\bar{b}}_{W}}{\gamma }} \end{array}$$(8)$$\begin{array}{c} \gamma^{2}{\bar{r}}_{W}-\frac{\bar{r}{\bar{b}}_{W}}{\bar{b}}=0 \end{array}$$

Solving (Equation 8) with respect to γ yields
(9)$$\gamma =\sqrt{\frac{\bar{r}{\bar{b}}_{W}}{{\bar{r}}_{W}\bar{b}}}$$

Along with γ, the actual error *E* between the neutral illumination color gamuts $\bar{r}$, $\bar{g}$, $\bar{b}$ and ${\bar{r}}_{W}$, ${\bar{g}}_{W}$, and ${\bar{b}}_{W}$; the expected error $E^{*}$ after scaling ${\bar{r}}_{W}$ and ${\bar{b}}_{W}$ by γ; and the updated value $K_{W}^{+}=K_{W}\Gamma_{K}$ are computed using (Equation 5):(10)$$\begin{array}{cc} E & ={\left(\bar{r}-{\bar{r}}_{W}\right)}^{2}+{\left(\bar{g}-{\bar{g}}_{W}\right)}^{2}+{\left(\bar{b}-{\bar{b}}_{W}\right)}^{2}\;\;\mathrm{and}\; \end{array}$$(11)$$\begin{array}{cc} E^{*} & ={\left(\bar{r}-{\bar{r}}_{W}\gamma \right)}^{2}+{\left(\bar{g}-{\bar{g}}_{W}\right)}^{2}+{\left(\bar{b}-\frac{{\bar{b}}_{W}}{\gamma }\right)}^{2} \end{array}$$

Based on these equations, the $K_{W}$ setting of the 3D DS camera is updated $K_{W}=K_{W}\Gamma_{K}$ if $E^{*}<E$. During testing, it was found that numerical inaccuracies can prevent the computation of the appropriate estimates for the color temperature *K* of the predominant illuminant. Therefore, a numerically stable test is used instead to determine whether $K_{W}$ has to be updated or its current value can be kept.
(12)$$\left(E>1.758\times 10^{-8}\right)\wedge \left(E-E^{*}>10^{-15}\right)$$

#### 2.2.2. Patient-Locked Auto-Exposure

In addition to the overall color appearance, the light sources that are present also affect the overall light intensity *I*, which among others, can vary depending on the viewing direction of the 3D DS camera. For example, in the case shown in Figure 3a, the camera is pointing toward a window. In Figure 3b, the camera is pointing in the opposite direction toward the door.

In order to maintain a constant illumination intensity *I* of the patient’s torso, independent of the viewing direction and the overall brightness of all present light sources, the histogram-based auto-exposure AE algorithm proposed in [24] was adopted.

This algorithm is implemented in the exposure component (Figure 1). It considers only the pixels in Ξ that correspond to the patient’s torso. These pixels are selected by segmenting the depth image *D* recorded by the 3D DS camera into a foreground object (the patient) and the remaining background using the approach outlined in Section 2.2.3. The binary mask $M_{D}$ obtained in this segmentation step is mapped to the color image Ξ using the texture coordinates $v_{uvs}$ computed from the depth image $D_{16}$ by the camera control library. All brightness values $I_{i}$ of all pixels covered by the mapped mask $M_{D}^{\prime }$ are considered for adjusting τ. Any other pixels and pixels that are over- or underexposed according to Equation (Equation 4) are discarded.

The algorithm proposed by Chen and Li [24] uses the histogram of the gamma-compressed grayscale image $\bar{\Xi }$ computed from Ξ. In order to avoid the computational burden required by an explicit conversion between the linear illumination image *I* and $\bar{\Xi }$, the histogram H(V) is directly computed from the linearized illumination values $I_{i}$ of the selected pixels. This is accomplished by maintaining a lookup table that lists the linearized bin boundary values ${\hat{h}}_{v}$ corresponding to the uniform boundaries $h_{v}$ of the grayscale histogram H(V). The histogram H(V) can then be generated for all considered $I_{i}$ using a left bisection search to scan this lookup table, which is far less computationally demanding. A further reduction can be achieved by precomputing the differences $\Delta_{H}^{2}={\left(I_{i}-128\right)}^{2}$ and $\Delta_{H}^{3}={\left(I_{i}-128\right)}^{3}$ for each bin, which are used to calculate the skewness S(V) of H(V).

To compute the values of the exposure time τ and overall gain $\Gamma_{ex}$ to be set on the camera, the overall exposure parameter $\overset{*}{\tau }$ is used.
$$\begin{array}{c} {\overset{*}{\tau }}_{n+1}={\overset{*}{\tau }}_{n}-\frac{S(V)}{\tau_{\Delta }}N_{\tau }\Gamma_{ex,n} \end{array}$$
(13)$$\begin{array}{c} \Gamma_{ex,n+1}=\max\left(\min\left(\frac{{\overset{*}{\tau }}_{n+1}}{\overset{**}{\tau }},\Gamma_{ex,max}\right),\Gamma_{ex,min}\right)\\ \tau =\max(\min\left(\frac{\overset{*}{\tau }}{\Gamma_{ex,n+1}},\tau_{max}\right),\tau_{min}) \end{array}$$
(14)$$\begin{array}{c} \overset{**}{\tau }=\max\left(\min\left(\frac{\tau_{frame}}{\tau_{\Delta }},\tau_{max}\right),\tau_{min}\right) \end{array}$$

The parameter $\tau_{\Delta }$ represents the size of one τ step in milliseconds, $N_{\tau }=5$ represents the number of steps to take if S(V)=1, and $\tau_{frame}=100\;\mathrm{ms}$ represents the optimal exposure time for each frame. The value of $\tau_{\Delta }$ depends on the actual step size in ms offered by the 3D DS camera.

#### 2.2.3. Depth Segmentation

The binary mask $M_{D}$ is created from the 16-bit depth images $D_{16}$ recorded by the 3D DS camera. It splits the image into the patient and any surrounding objects, obstacles, and relevant edges. This implementation was inspired by the Canny edge detection algorithm proposed in [25]. The algorithm uses two thresholds to find the edges in an image Ξ based on the gradient $\Delta \bar{\Xi }$ of its corresponding grayscale image $\bar{\Xi }$. Pixels that have a gradient value $\Delta x_{i}$ that exceeds the upper limit are considered to be part of an edge. Pixels with a value of $\Delta x_{i}$ between the two limits are only included in an edge if they are adjacent to an already identified edge pixel. To improve the obtained set of edges and reduce the number of edges caused by noise, the grayscale $\bar{\Xi }$ is smoothed using a Gaussian filter.

This approach was adopted for processing depth images $D_{16}$ that contain pixels for which no valid depth value $D_{i}=0$ is available. The computation of the depth value gradient $\Delta D_{i}$ and one of the corresponding Gaussian filter weights $w_{i}$ are computationally too demanding to be computed in real time. Therefore, the depth gradient values $\Delta D_{i}=\sqrt{\Delta d_{x,i}^{2}+\Delta D_{y,i}^{2}}$ of *D* are rounded to the closest 16-bit integer value $\Delta_{16}D$. The resulting reduced number of $\Delta_{16}D$ and corresponding distinct weights $w_{16}$ are stored in a precomputed weights table instead of directly computing $w_{i}$ on every iteration for each pixel. This avoids the computationally demanding operations of computing $e^{x}$ and $\sqrt{x}$ in real time. A companion table with squared boundary values $\Delta_{16}^{2}D=\left(\Delta_{j,16}^{2}D+\Delta_{j+1,16}^{2}D\right)\;/\;4$ between the individual $\Delta_{16}D$ ensures that the $w_{i}$ for a pixel $D_{i}$ of *D* can be generated through a fast left bisection search. Pixels $D_{i}=0$ represent objects without a defined depth, and their values are copied to the smoothed depth image $\tilde{D}$ image without any changes.

The smoothed $\tilde{D}$ is filtered using an octagonal Laplace kernel to find the initial set of edge pixels $d_{e}$,
(15)$$\left[\begin{array}{ccc} -\frac{1}{\sqrt{2}} & -1 & -\frac{1}{\sqrt{2}}\\ -1 & 4+2\sqrt{2} & -1\\ -\frac{1}{\sqrt{2}} & -1 & -\frac{1}{\sqrt{2}} \end{array}\right]$$

An octagonal kernel has the advantage that all distances between the eight-connected neighbor pixels $d_{8}$ and the central pixel $d_{c}$ are of equal length.

All pixels *d* that exhibit a sign change between opposing neighbor pixels $\nabla d_{8}$ on the Laplacian image $\nabla \tilde{D}$ are included in the initial set of edge points $d_{e}$. Pixels $d_{e}$ that have at least one neighbor $d_{k,8}=0$ with an undefined depth are considered primary edge pixels $e_{P}$. Their actual $\Delta \tilde{D}\left(d_{e}\right)$ values are computed using the following approach:(16)$$\Delta \tilde{D}\left(d_{e}\right)=\max\left(\left\{\begin{array}{cc} {\tilde{D}}_{+k,8}-{\tilde{D}}_{-k,8} & \mathrm{if}\;{\tilde{D}}_{-k,8}>0\;\mathrm{and}\;{\tilde{D}}_{+k,8}>0\\ 2*{\tilde{D}}_{-k,8}-{\tilde{D}}_{i} & \mathrm{if}\;{\tilde{D}}_{-k,8}>0\\ 2*{\tilde{D}}_{+k,8}-{\tilde{D}}_{i} & \mathrm{if}\;{\tilde{D}}_{+k,8}>0 \end{array}\right.\right)$$

All $d_{e}$ where $\Delta \tilde{D}\left(d_{e}\right)>\Delta {\tilde{D}}_{P}$ are marked $d_{P}$, whereas any other $d_{e}$ are only considered if the Canny rule for minor edge pixels $d_{M}$ holds. This rule has been modified for use on depth images *D* as follows:(17)$$\begin{array}{c} \left(\Delta \tilde{D}\left(d_{e}\right)>\Delta {\tilde{D}}_{M}\right)\wedge \left(\left(\Delta \tilde{D}\left(d_{e}\right)>\Delta {\tilde{D}}_{P}\right)\vee \left(\frac{\Delta \tilde{D}\left(d_{e}\right)-\Delta {\tilde{D}}_{M}}{\Delta {\tilde{D}}_{P}-\Delta \tilde{D}\left(d_{e}\right)}>1\right)\right) \end{array}$$

The upper Canny limit $\Delta {\tilde{D}}_{P}$ is set to 1.2 cm and the minor limit $\Delta {\tilde{D}}_{M}$ is set to 0.35 cm.

A binary depth mask $M_{D}$ is created from all pixels $d_{i}>0$ in *D* of a known depth. Pixels $d_{e}$ located at any of the edges are excluded from $M_{D}$. The resulting $M_{D}$ is split into 9 segments $M_{D,9}$. The pixels $M_{i}$ within the central $M_{D,9}$ are labeled with respect to the different objects and components they represent. The labeled four-connected components $L_{4}$ are sorted by size. The largest $L_{4}$ that touches the segment boundary is extended to all other $M_{D,9}$ segments using the flood-fill method, starting from the center of mass of $L_{4}$. At the end of this step, all adjacent edge pixels $d_{e}$ are appended to the extended $L_{4}^{+}$ representation of $L_{4}$.

As the depth values at the boundaries of $L_{4}^{+}$ can largely vary, the following approach is used to remove any unrelated outliers. This approach is based on the observation that the boundaries of the patient’s torso are well-separated from the background along the vertical direction and above the head.
(18)$$\begin{array}{c} D_{close}<D_{i}\left(M_{i}\right)<D_{far} \end{array}$$
(19)$$\begin{array}{c} D_{far}=\min\left(\max\left(D_{r,far}\right)+\frac{\max\left(D_{r,far}\right)-\min\left(D_{r,close}\right)}{3},\;\bar{D_{r,far}}+3\sigma \left(D_{r,far}\right)\right) \end{array}$$
(20)$$\begin{array}{c} D_{close}=\max\left(\min\left(D_{r,close}\right)-\frac{\max\left(D_{r,close}\right)-\min\left(D_{r,close}\right)}{3},\;\bar{D_{r,close}}-3\sigma \left(D_{r,far}\right),\;D_{near}\right) \end{array}$$

The values $D_{r,close}$ and $D_{r,far}$ correspond to the smallest and largest depth values encountered for the mask pixels $M_{i}$ within each row of $L_{4}^{+}$, and $\bar{D_{r,close}}$, $\bar{D_{r,far}}$, $\sigma D_{r,close}$, and $\sigma (D_{r,far})$ represent their mean and standard deviations. Any pixel $M_{i}$ for which the condition in (Equation 18) does not hold is removed from $L_{4}^{+}$. In the case that either the number of pixels of $L_{4}^{+}$ is less than 200 or no appropriate values for $D_{far}$ or $D_{close}$ could be found, the current $L_{4}$ is discarded and the search for a suitable $L_{4}^{+}$ representing the patient is attempted with the next larger $L_{4}$. If no suitable $L_{4}$ is left, segmentation is aborted and real-time processing continues with the next set of depth and color image frames recorded by the 3D DS camera.

#### 2.2.4. Surface Mesh Generation

The final surface mesh is generated by converting the depth image *D* into a corresponding point cloud *P*. Therein, each point $v_{i}$ corresponds to a specific pixel $d_{i}$ in *D*. In the case of pixels $d_{i}=0$ without a defined depth value, the origin point $v_{i}=O=\left(0,\;0,\;0\right)$ is assigned. The unique correspondence between any $d_{i}$ and its corresponding $v_{i}$ allows creating *S* by mapping a pre-triangulated grid *G* to *P*. Any triangle *T* that includes at least one $v_{i}$ for which $d_{i}=0$ is dropped from *G*.

Before *S* is stored on disk using the **.obj** format, along with $\Xi_{uvs}$ and the color temperature setting $K_{W}$ it was recorded with, degenerated $T_{A=0}$ and occluded triangles $T_{-1}$ that do not correspond to a valid surface patch are removed. The filtering of $T_{-1}$ is facilitated by the fact that 3D DS cameras, especially those that can capture objects located a short distance from the camera, use a dedicated RGB color sensor to record $\Xi_{uvs}$. This sensor is typically attached to the left or right side of the depth sensor system and thus views the imaged object from a slightly different angle. This difference in viewing angle and FOV between the depth and the color sensor is sufficiently large to identify triangles that do not represent a part of the object’s real surface. This small difference in viewing angle causes the surface normal $n_{-1}$ to flip its direction between the representation of $T_{-1}$ in the depth image *D* and in Ξ. This flip is not plausible as it would mean that the color sensor is capturing the back side of $T_{-1}$, whereas the depth sensor captures its front side. This is prevented by the fact that both sensors are mounted on the same support. The following approach exploits this fact by identifying triangles where the sign, and thus the direction, of the surface normal vector appears flipped in Ξ compared to *D*.

The pre-triangulated grid *G* is initialized such that the normal vector $n_{T}$ of each triangle *T* on *S* points toward the camera and is oriented in the negative 〈m,Z〉<0 viewing direction Z of the camera. For every valid *T* of initial surface mesh *S*, the normal vector $n_{uvs}$ of its representation in Ξuvs$T_{uvs}$ must also point in the $-Z_{uvs}$ direction. Triangles $T_{-1}$ where the signs of n and $n_{uvs}$ are opposite, indicated by $\langle n_{uvs},\;Z_{uvs}\rangle >=0$, suggest that triangle $T_{-1}$ likely does not represent a valid part of *S* and should be removed.

In addition, triangles $T_{A=0}$ with a degenerated representation $T_{uvs}$ in $\Xi_{uvs}$ are removed. This includes triangles with an area $A_{uvs}<0.25$ pixels, as well as cases where $T_{uvs}$ has a shortest edge of less than half a pixel and triangles that extend beyond the top and bottom corners of $\Xi_{uvs}$.

Further, skinny triangles $T_{\varphi <13}$ are discarded if they enclose at least one angle φ between any two edges $e_{a}$, $e_{b}$, and $e_{c}$ that is smaller than 13 degrees, and if the lengths $|e_{c}|$ and $|e_{b}|$ of its longest two edges $e_{c}$ and $e_{b}$ conform to the following conditions: (21)$$\begin{array}{c} \left(\right|e_{c}\left|>\bar{|e_{KNN}|}+4\sigma |e_{KNN}|\right)\wedge \left(e_{b}|>\bar{|e_{KNN}|}+4\sigma |e_{KNN}|\right) \end{array}$$(22)$$\begin{array}{c} (|e_{c}|>\bar{|e_{KNN}|}+4\sigma |e_{KNN}|)\wedge \left(\mathrm{count}\left(\varphi <13\right)==2\right) \end{array}$$

To compute the average length $\bar{|e_{KNN}|}$ and standard deviation $\sigma |e_{KNN}|$, only triangles $T_{KNN}$ are considered that are formed by any three K-nearest neighbors $v_{KNN}$ located within a radius of $\max(|e_{b}|*0.9,\;|e_{a}|)$ around the tip vertex $v_{\varphi <13}$ of $T_{\varphi <13}$ and the midpoint of its shortest edge $e_{a}$. Additionally, any $T_{\varphi <13}$ that has to be discarded according to (Equation 21) will result in the deletion of all adjacent $T_{\varphi <13}$ connected to its $e_{b}$ or $e_{c}$. In addition, in the case of any $T_{\varphi <13}$ satisfying (22), only the $T_{\varphi <13}$ adjacent to $e_{c}$ is removed. Finally, duplicate ${\mathrm{vi}}_{i}\equiv v_{j}$ encoding the same point and v not referenced by any triangle are removed from the surface *S*, along with all small disconnected surface patches $S_{dis}$.

The surface *S* is stored on disk in **.obj** format, along with the corresponding texture information $\Xi_{uvs}$. Its triangle $n_{T}$ and vertex normals $n_{v}$ are recomputed, and a transformation ***ℜ*** is applied to all vertices and normals. The latter ensures that the z-axis points in the direction of the patient’s head and the positive x-axis extends from the left to the right side of the torso. The origin point is selected such that it is located on the central viewing axis of the camera. To compute its y-component, the point cloud is divided into 3 sections along the vertical direction, roughly representing the chest, belly, and hips of the patient from top to bottom. The points within the top third are further split into 5 subgroups from right to left along the x-axis. For the rightmost and leftmost groups, the median coordinates ${\hat{y}}_{r}$ and ${\hat{y}}_{l}$ are computed. Based on these values, the final y-coordinate of the origin point is computed as $y={\hat{y}}_{r}+{\hat{y}}_{l}\;/\;2$.

This ensures that all surfaces are located close to each other and that they partially overlap. At the same time, the actual relative shift between the surfaces and the angle at which the camera views the surface is retained as much as possible. This is crucial for the registration process described in Section 2.3.2.

### 2.3. Offline Processing

The electrode positions are computed using a set of at least 14 recordings of the torso surface, covering a minimum angle of approximately ≈270 degrees in the horizontal plane. The necessary steps, depicted in Figure 1, are presented in the following subsections. These steps include the pairwise alignment and registration of the recorded surfaces *S*, as described in Section 2.3.2; the extraction of the points v representing the colored electrode markers, as described in Section 2.3.3; and the fitting of a model of the marker to identify its central point, as described in Section 2.3.4. In the final step, a unique label is attached to each position, which uniquely links the individual ECG signals and the 3D position of the corresponding electrode.

#### 2.3.1. Color Correction

The color sensor of the Intel Realsense SR300 camera (Intel corporation, Santa Clara, CA, USA) offers only a limited range (between $K_{W,min}=2500$ and $K_{W,max}=6500$) within which the color temperature parameter $K_{W}$ can be tuned using the algorithm discussed in Section 2.2.1. This range is optimized for indoor use [21,22], where typical light sources include incandescent tungsten lamps (K=2500), fluorescent lights (K=3800), and standardized CIE sources such as CIE55 (K=5000) or CIE65 (K=6500).

The space limitations encountered in clinical settings, for example, outpatient and cardiology practitioner clinics, result in more challenging illumination conditions that can vary significantly depending on factors such as the patient’s seating position or the camera’s direction. Specifically, individual objects and parts of the room may be shaded by other objects, for example, the electrodes on the patient’s back. Shaded areas are characterized by color temperature values K>7000, which are significantly larger than the $K_{W,max}=6500$ upper limit assumed by the color sensor. Examples of this situation are shown in Figure 4a,c.

An additional color-correction process is applied to the recorded texture images Ξ and the 3D surfaces. A virtual camera is used to simulate the recording of Ξ with a different $K_{W}$ setting than the actual one. This virtual camera offers an AWB range between $K_{W,min}=2000$ and $K_{W,max}=9000$. It uses the model introduced in Section 2.2.1 to adjust the gain of its red $\Gamma_{\hat{R}}=\Gamma_{K}$ and blue $\Gamma_{\hat{B}}=1/\Gamma_{K}$ color channels.

The virtual camera internally stores a linearized and normalized representation ${\hat{\Xi }}_{=}$ of $\Xi_{uvs}$. This representation corresponds to an image recorded with an equal gain $\Gamma_{K}=1$ and $K_{W}=5500$.
(23)$$\begin{array}{c} \Gamma_{K,=}=\frac{2(K_{W,uvs}-K_{W,min})}{K_{W,max}-K_{W,min}},\;{\hat{R}}_{i,=}=\frac{{\hat{R}}_{i}}{\Gamma_{K,=}},\;{\hat{G}}_{i,=}={\hat{G}}_{i},\;{\hat{B}}_{i,=}={\hat{B}}_{i}\Gamma_{K,=},\;\Gamma_{K}=\Gamma_{K,=} \end{array}$$

Its white-balancing parameter $K_{W}$ is initialized to the color temperature $K_{uvs}$ at which ${\hat{\Xi }}_{uvs}$ was recorded by the color sensor of the 3D DS camera.

After initialization, the color-correction approach described in Section 2.2.1 is used to adjust the $K_{W}$ of the virtual camera until a suitable value for $K_{W}^{+}$ is found. If $K_{W}^{+}$ jitters around its ideal value for at least 20 repetitions, the color correction stops when the following condition is met:(24)$$-1<=K_{n+1,W}^{+}-K_{n,W}^{+}<=1$$

In this case, $K_{W}^{+}$ is set to the mean value $\bar{K_{W}^{+}}$ of the last 3 minimum updates for which the difference between consecutive $K_{W}^{+}$ values is less than 10. With each update of $K_{W}$, a new version of $\Xi_{uvs}$ is created by multiplying the red color values ${\hat{R}}_{=}$ of ${\hat{\Xi }}_{=}$ by the updated $\Gamma_{K}^{+}$, multiplying the blue values by $1\;/\;\Gamma_{K}^{+}$, and performing a left bisection search on the lookup table $\hat{V}=\left\lfloor V\right\rfloor$ established in Section 2.2.1. Pixels that are overexposed according to (4) are not modified. Pixels that appear overexposed after scaling and exceed a maximum value of 1 in at least one channel are assumed to be fully saturated in all three channels, which are each set to the maximum value. Pixels that appear underexposed, with at least one channel having a value less than $10^{-8}$, are assumed to be unexposed in all channels. Therefore, in such cases, all three channels of the pixel are set to 1 when fully saturated and 0 when unexposed. Additionally, all channels are clipped to the maximum possible value of 1 if necessary. The color-optimized version of $\Xi_{uvs}$ (Figure 4b,d) is then used to extract the 3D points of the electrode markers, as described in Section 2.3.3.

#### 2.3.2. Surface Registration

To align the surfaces, a point-to-plane algorithm was chosen. This kind of ICP algorithm minimizes the distances $l=|ℜv_{S}-v_{T}|$ between corresponding $v_{S}$ and $v_{T}$ along the direction of the surface normals $n_{T}$ of $S_{T}$.
(25)$$E\left(ℜ\right)=\sum_{(v_{S},v_{T})\in K}{∥\left(v_{T}-ℜv_{S}\right)n_{T}∥}^{2}=min$$

A precise alignment between $S_{T}$ and $S_{S}$ across all surface pairs is achieved when Equation (Equation 25) is also minimal in the reverse case with $S_{T}$ and $S_{S}$ swapped. The following simple symmetric point-to-plane approach is used by the registration component (Figure 1) to align the surfaces. It was chosen in favor of other symmetric point-to-plane algorithms such as [26], as it can be directly implemented using unidirectional ICP functions from open3D library [27]. In the first step, the forward transformation matrix ${ℜ}_{f}$ is computed for the set of corresponding points $\left(v_{T},v_{S}\right)\in C_{f}$ by applying (Equation 25). In the second step, the reverse transformation ${ℜ}_{R}$ is computed for the points $\left(v_{S},v_{T}\right)\in C_{r}$ corresponding to the reversed setup. The initial ${ℜ}_{0,r}$ is initialized as ${ℜ}^{-}1_{f}$. The set $\left(v_{T},v_{S}\right)\in C_{f}$ is selected from a subset of $v_{S}$ that is located within the maximum correspondence distance $l_{c}$ of $v_{T}$. The same selection criterion is used for the reverse set $\left(v_{S},v_{T}\right)\in C_{r}$ with respect to any $v_{S}$. In the final step, the optimal transformation ***ℜ*** and the new correspondence distance $l_{c,+1}$ are selected from ${ℜ}_{f}$, ${ℜ}_{r}$, $l_{c+1,f}$, and $l_{c+1,r}$ using the following criteria:(26)ℜ,lc+1=ℜr,lc+1,rifE(ℜr)<{E(ℜ),E(ℜf)}∧lc+1,r<{lc,lc+1,f}ℜf,lc+1,fifE(ℜf)<{E(ℜ),E(Rer)}∧lc+1,f<{lc,lc+1,r}lc+1,f=l¯f+2σlf,lc+1,r=l¯r+2σlrl¯f=mean(vT,vS)∈Kf(|vS−vT|),σlf=std(vT,vS)∈Cf(|vS−vT|)l¯r=mean(vS,vT)∈Cf(|vT−vS|),σlr=std(vS,vT)∈Cr(|vT−vS|)

The surfaces *S* recorded using the approach described in Section 2.2 are aligned such that they more or less share the same space, apart from the small rotation Δφ along the horizontal direction and the relative vertical movement Δz between the cameras. No information about their orientation in space or how much each pair overlaps is recorded. For obtaining sufficiently precise positions of the electrodes, the optimal correspondence distance $l_{o}$ between any $\left(v_{T},v_{S}\right)$ should be $l_{o}≲1\;\mathrm{mm}$. Therefore, the symmetric ICP registration is repeated for each pair in multiple runs. The results obtained for ***ℜ*** and $l_{c+1}$ in the previous run are used to initialize ${ℜ}_{0}$ and $l_{c}$ in the next run. If the condition in (Equation 26) for updating ***ℜ*** and $l_{c}$ fails, one last run is attempted with $l_{c}=l_{min}\approx 1\;\mathrm{mm}$ if $l_{c}<l_{min}$ and $l_{c-1}-l_{c}>\sigma l_{0}$ holds. For the first optimization run, ${ℜ}_{0}$ is initialized to roughly reflect the relative rotation about the z-axis between two recorded surfaces $S_{T}$ and $S_{S}$ and its relative shift Δz along the z-axis. The following approach is used to estimate the relative rotation angle Δφ between $S_{T}$ and $S_{S}$:(27)$$\Delta \varphi =\arccos\left(\frac{\langle {\hat{v}}_{l,S}-{\hat{v}}_{r,S}|{\hat{v}}_{l,T}-{\hat{v}}_{r,T}\rangle }{|h_{S}\left|\right|h_{T}|}\right)$$

The right ${\hat{v}}_{r,S}$, ${\hat{v}}_{r,T}$ and left ${\hat{v}}_{l,T}$, ${\hat{v}}_{r,T}$ median points define the horizontal directions of the sagittal planes with respect to $S_{S}$ and $S_{T}$. They are computed using the same approach described in Section 2.2.4 to define the final position of the origin along the y-coordinate.

Suitable estimates for $l_{c,max}$, $l_{c,min}$, and $\sigma l_{0}$ are essential for achieving a sufficiently precise alignment of $S_{T}$ and $S_{S}$. When testing the implementation of the symmetric ICP, it was empirically found that the values for $l_{c,max}$, in particular, varied significantly depending on the relative distance and angle between two consecutive surfaces. Initially, constant values were assigned to $l_{c,max}$ and $l_{c,min}$. However, these values resulted in an insufficient alignment between the surfaces on average. Specifically, the alignment of the surfaces at the left side where the front and back sides of the torso meet was rather challenging, and in some cases, not possible at all.

In order to improve the results and ensure a proper alignment between the surfaces, the following approach is used to determine suitable estimates of $l_{c,max}$, $l_{c,min}$, and $\sigma l_{0}$ for each pair of $S_{T}$ and $S_{S}$. These estimates are computed based on the distances between the vertices $v_{T}$ and $v_{S}^{\prime }$ within the volume $V_{T\cap S^{\prime }}=V_{T}\cap V_{S}^{\prime }$, which represents the common region of the axis-aligned bounding boxes $V_{T}$ and $V_{S}^{\prime }$ encompassing the target surface $S_{T}$ and the source surface $S_{S}^{\prime }$. The latter $S_{S}^{\prime }$ is obtained by applying an initial transformation ${ℜ}_{0}$ to the source surface $S_{S}$. The transformation ${ℜ}_{0}$ shifts all $v_{S}^{\prime }\in V_{T\cap S^{\prime }}$ such that their center of mass $\hat{v_{S}^{\prime }}$ aligns with the center of mass $\hat{v_{T}}$ of all $v_{S}\in V_{T\cap S^{\prime }}$. The value for $l_{max}$ is obtained by applying (Equation 26) to the distances between the points in the forward correspondence set $\left(v_{T},v_{S}^{\prime }\right)\in C_{f,0}$ and the backward correspondence set $\left(v_{T},v_{S}^{\prime }\right)\in C_{r,0}$. Both sets are found through a KNN search [28,29], which also considers the surface normals $n_{T}$ and $n_{S}^{\prime }$ in each $v_{T}$ and $v_{S}^{\prime }$. This approach has the advantage of considering only $v_{T}$ and $v_{S}^{\prime }$ as corresponding when their surface normals $n_{T}$ and $n_{S}^{\prime }$ are closely aligned. From the resulting $C_{f,0}$ and $C_{r,0}$, any $v_{T}$ and $v_{S}^{\prime }$ are removed if the deviation between their surface normals $n_{S}^{\prime }$ and $n_{T}$ exceeds 30 degrees, ensuring that $\langle n_{S}^{\prime }|n_{T}\rangle <0.98$.

The estimate for $l_{min}$ is based on the overall $\mathrm{mean}(|v_{T}|,|v_{S}^{\prime }|)$ of the shortest neighbor distances within all $v_{T}\in V_{T\cap S^{\prime }}$ and $v_{S}^{\prime }\in V_{T\cap S^{\prime }}$.

From the final ***ℜ*** of all consecutive pairs of $S_{T}$ and $S_{S}$, the global alignment of each $S_{i}$ is determined by the cumulative transformation $ℜ=\prod_{j}^{i}{ℜ}_{j}$, starting with identity Re=I for the first surface $S_{1}$. Alternatively, the transformation of the first surface can be initialized by the horizontal camera inclination angle φ about the z-axis using (Equation 27). From this, the relative angle between $S_{1}$ and the x-axis of the patient’s frontal plane is computed. This already provides a rough alignment of the resulting point cloud of the torso with its frontal plane.

#### 2.3.3. Electrode Marker Extraction

In the current setup, the electrodes are attached to g.LADYbird^TM^ active electrode clips from g.tec medical engineering GmbH, Schiedlberg, Austria. These clips have a circular head, with its center aligned with the center of the electrode. The clip itself is covered with red-colored epoxy to protect the integrated electronics from water and other liquids. The circumference of the head is painted blue to model a circular electrode marker with a blue boundary and a red central disk. Figure 5 shows an example of this basic setup.

The blue boundary color (see Figure 5b) is selected such that the electrode marker easily can be detected within the RGB chromacity space representations ${\hat{\chi }}_{rg}$ of the surface texture images $\Xi_{uvs}$. The ${\hat{\chi }}_{rg}$ values are obtained as a byproduct of the white-balancing and light color–temperature correction approaches described in Section 2.3.1.

Each ${\hat{\chi }}_{rg}$ is scanned for red $x_{r}=r_{r},g_{r},b_{r}$ and blue $x_{b}=r_{b},g_{b},b_{b}$ pixels that are fully described by one of the following two ellipses within the RGB chromacity space.
(28)$$\begin{array}{c} \frac{{\left(\delta r_{r}\cos\left(\phi_{r}\right)-\delta g_{r}\sin\left(\phi_{r}\right)\right)}^{2}}{\sigma (r_{r})}+\frac{{\left(\delta r_{r}\sin\left(\phi_{r}\right)+\delta g_{r}\cos\left(\phi_{r}\right)\right)}^{2}}{\sigma (g_{r})}<=1\\ \mathrm{with}\;\delta r_{r}=r_{r}-\bar{r_{r}},\;\delta g_{r}=g_{r}-\bar{g_{r}} \end{array}$$
(29)$$\begin{array}{c} \frac{{\left(\delta r_{b}\cos\left(\phi_{b}\right)-\delta g_{b}\sin\left(\phi_{b}\right)\right)}^{2}}{\sigma (r_{b})}+\frac{{\left(\delta r_{b}\sin\left(\phi_{b}\right)+\delta g_{b}\cos\left(\phi_{b}\right)\right)}^{2}}{\sigma (g_{b})}<=1\\ \mathrm{with}\;\delta r_{b}=r_{b}-\bar{r_{b}},\;\delta g_{b}=g_{b}-\bar{g_{b}} \end{array}$$

The values $\bar{r_{r}}$, $\bar{g_{r}}$, $\bar{r_{b}}$, $\phi_{r}$, $\bar{g_{b}}$, and $\phi_{b}$ define the red and green coordinates of the center point of the ellipsis and the rotation angle by which each of them is rotated with respect to the red axis of the RGB chromacity space. Their values are determined through the calibration procedure described in Section 2.3.6. All matching $x_{r}$ and $x_{b}$ pixels are mapped to their corresponding 3D vertices $v_{r}$ and $v_{b}$ on the torso surface *S*. This mapping is accomplished by computing the barycentric coordinates of each $x_{r}$ and $x_{b}$ within the representation of the surface triangle *T* in $\Xi_{uvs}$.

The resulting marker point cloud $P_{M}$ formed by all $v_{r}$ and $v_{b}$ is filtered with respect to $v_{r}$ and $v_{b}$, which likely correspond to a valid electrode marker, as defined by the color of the clip head. This is achieved by a radius-based KNN search for at least one neighbor of the opposite color. The radius is set to the radius of the clip head for all $v_{r}$ and the width of the blue boundary ring for all $v_{b}$. If the neighborhood of radius ρ does not contain any points of the opposite color, v is removed from $P_{M}$.

The filtered $P_{M}$ is split into individual clusters of $v_{el}\in v_{r}\cup v_{b}$, representing the individual electrode clips. This is accomplished by applying the HDBSCAN algorithm [30]. The results are more robust compared to the basic DBSCAN algorithm [31], especially in the presence of groups of outliers, for example, generated by a bluish shadow cast on the cables and electrode clips. In addition, a minimum distance $\epsilon_{split}$ can be defined, and clusters are not split any further. In contrast to the basic DBSCAN [31] algorithm, $\epsilon_{split}$ defines a lower boundary limit rather than a strict cutting distance. In other words, less dense clusters with an average density exceeding $\epsilon_{split}$ are not necessarily forced to split into distinct leaf clusters. The parameters of the minimum cluster size $N_{C,min}$ and the minimum samples $X_{C,min}=20$ are used to fine-tune and control the extraction of clusters that represent the individual electrode markers, considering the actual number of electrodes $N_{el}$.
(30)$$\begin{array}{c} \epsilon_{split}=r_{nh} \end{array}$$
(31)$$\begin{array}{c} N_{C,min}=\max\left(\frac{\mathrm{count}(v_{r})}{4N_{clip}},X_{C,min}\right) \end{array}$$

In order to simplify the subsequent processing steps, the overall point cloud $P_{S}$, as well as PM, is realigned such that the frontal plane of the torso is in line with the x-z plane of the coordinate system. This is achieved by once again splitting $P_{S}$ into chest, belly, and hip sections. The points of the chest section are further split along the x-axis into three parts, representing the right shoulder, neck, and left shoulder. The final transformation ***ℜ*** is computed by aligning the vector between the median points of the left and right shoulders to the x-axis of the frontal plane.

#### 2.3.4. Fitting Marker Model

The red points $v_{r}$ and blue points $v_{b}$ within each cluster are fitted to a planar marker model consisting of a red disk enclosed within a blue ring. Before fitting, all $v_{r}$ and $v_{b}$ are projected onto the plane $Q_{cl}$, which is parallel to all $v_{el}$.
(32)$$v_{e}^{\prime }l=v_{el}-\langle v_{el}-{\bar{v}}_{el},\;n_{mj}\rangle n_{mj}$$

This ensures that all $v_{el}$ are located on $Q_{cl}$, which is defined by the predominant surface normal vector direction $n_{mj}$ within all surface normal vectors $n_{el}$$v_{el}$ and their center of mass ${\bar{v}}_{el}$.
(33)$$\begin{array}{c} n_{mj}=\overset{r}{\sum_{i}}U_{i}\sigma_{i}V_{i}\;\mathrm{with}\;U\Sigma V=\mathrm{svd}\left(n_{G}\left(v_{el}\right)\right)\\ r=i_{max}\underset{1\leq i\leq K}{\Rightarrow }-10\log_{10}\left(\frac{\sum_{j}^{i}\sigma_{j}}{\sum_{k}^{K}\sigma_{k}}\right)>3 \end{array}$$

The shifted $v_{el}^{\prime }$ are then fitted to the following model, which is based on the distances $\rho (v_{el}^{\prime })$ between the individual $v_{el}^{\prime })$ and the electrode center X on $Q_{el}$.
(34)$$\begin{array}{c} \rho \left(v_{el}^{\prime }\right)=|v_{el}^{\prime }-X\left|,\;\delta_{\rho }\left(b_{b}\right)=\rho \left(v_{b}\right)-\rho_{disc}\right)\\ \epsilon =\sum \delta \rho^{2}\left(v_{b}\right)+\sum \rho^{2}\left(\left\{v_{r}^{\prime }|\rho \left(v_{r}^{\prime }\right)<\rho_{disc}\right\}\right)+\sum \rho^{2}\left(v_{b}^{\prime }\right)+{\langle X-{\bar{v}}_{el},n_{mj}\rangle }^{2} \end{array}$$

$\delta_{\rho }$ represents the relative distances of the blue points $v_{b}^{\prime }$ from the boundary of the enclosed red disk with a radius $\rho_{disc}$. From all the red points $v_{r}$ within a cluster, the model selects those that are within a radius $\rho_{r}<\rho_{disc}$ from the current X. The model in (Equation 34) is optimized with respect to X using the L-BFGS-B algorithm provided by the SciPy minimize function. This numerically robust algorithm was selected because it can achieve satisfactory optimization results for least-squares optimization problems. Its implementation details can be found in the SciPy manual and [32,33]. For all clusters for which an $\epsilon_{min}\left(X\right)$ could be found, $X_{min}$ is stored, along with $n_{mj}$. Any remaining clusters for which no appropriate $X_{min}$ could be found are not further considered.

In some cases, it is possible that a clip is split into two smaller clusters. For example, if an electrode array is carelessly attached to the torso, electrode leads can shadow the relevant parts of the clip head. This might be the case when the following condition holds with respect to the counts of $v_{r}$ or $v_{b}$:(35)$$\frac{1}{3}<\frac{\mathrm{count}(v_{r})}{\mathrm{count}(v_{b})}<3$$

Two neighboring clusters are considered pieces of the same marker only if at least 10 closest neighbors of any $v_{el}$ in the first cluster are closest to at least 85 distinct $v_{el}$ in the other cluster. The cylindrical model is fitted to the largest piece of the marker only. This prevents nearby image artifacts in the $\Xi_{uvs}$ from causing misalignment of the affected electrode marker and distracting the center point from its true location.

The identified cluster centers X are triangulated using the ball-pivoting method [34,35] implemented in the open3D library. The radii $\rho_{1}=\bar{x}/\sqrt{2}$ and $\rho_{2}=\bar{x}$ for two distinct balls are derived from the average distance $\bar{x}=\mathrm{mean}(|X_{9}-X\left|\right)$ between each X and its 9 closest neighbors. Outliers are removed if $|X_{9}-X|>\bar{x}+2*\sigma (|X_{9}-X\left|\right)$ before computing $\bar{x}$. For a final check to determine if the X of neighboring clusters resemble two pieces of the same marker, the surface connectivity between individual X is computed. The marker attached to the largest group, where two X for which |X1−X2|<23ρ0 holds, is retained, whereas the other is removed. Ball-pivoting triangulation and the removal of small clip pieces are repeated until no more nearby groups, represented by distinct X, are found. The remaining X that are included in the resulting triangular surfaces represent the frontal and dorsal patches of the electrode grid layout proposed in [36]. Clusters that are too far away to be included in the mesh by the ball-pivoting process are considered single electrodes, similar to those used, for example, in Einthoven I, II, and III.

In the final step, the triangular meshes of the frontal and dorsal electrode patches are normalized. In this process, any vertical edge that intersects the horizontal line between two common neighbors of its endpoints is swapped with the edge that connects the common neighbors.

#### 2.3.5. Label Assignment

Starting from the point with the smallest y-coordinate, the triangulation of the frontal patch is scanned line by line. All electrodes that can be connected along consecutive horizontal edges are joined into one row of the frontal patch [36] and stored in right-to-left order. The rows are ordered from bottom to top. After all rows of the frontal patch have been collected, the same approach is applied to collect the electrodes of the dorsal patch. Again, the electrodes are stored in right-to-left and bottom-to-top order.

On the frontal patch, the number labels for each channel are assigned in ascending order from bottom right to top left. The dorsal assignment starts at the top right and ends at the bottom left. The remaining electrode points X that have not been included within the triangulation of the frontal and dorsal patches either correspond to the three Einthoven leads I, II, and III if they are located on the arms close to the front of the left and right shoulders and on the left hip. The electrode array includes two additional electrodes that are placed frontal and dorsal close to the right side of the torso.

#### 2.3.6. Calibration

The proposed method to identify the color electrode markers requires proper calibration of the mean values $\bar{r_{r}}$, $\bar{g_{r}}$, $\bar{r_{b}}$, and $\bar{g_{b}}$; the standard deviations $\sigma (r_{r})$, $\sigma (g_{r})$, $\sigma (r_{b})$, and $\sigma (g_{b})$; and the rotation angles $\phi_{r}$ and $\phi_{b}$ of the ellipses in Equations (Equation 28) and (Equation 29). In the first step, the color-corrected chromacity representation ${\hat{\chi }}_{rg}$ of the texture images $\Xi_{uvs}$ obtained as a byproduct in Section 2.3.1) is roughly segmented. The pixels representing a blue or red pixel of the clips are initialized with the following values: $\bar{r_{r}}=0.75$, $\sigma (r_{r})=0.1$, $\bar{g_{r}}=0.08$, $\sigma (g_{r})=0.06$, $\bar{r_{b}}=0.05$, $\sigma (r_{b})=0.02$, $\bar{g_{b}}=0.13$, $\sigma (g_{b})=0.06$, and $\phi_{r}=\phi_{b}=0$.

These values were empirically identified from the chromacity space triangle of the 3D DS camera’s color sensor, generated from the pixels of all ${\hat{\chi }}_{rg}$. The resulting raw pixel masks $M_{\hat{\chi },raw}$ are stored along with the corresponding ${\hat{\chi }}_{rg}$ obtained from the data sets of at least three patients. In addition, a binary mask $M_{I}$ selects pixels of $\Xi_{uvs}$ that are properly exposed according to (Equation 4). For storing the ${\hat{chi}}_{rg}$ on disk, the 16-bit PNG format is used. They are loaded along with the corresponding $M_{\hat{\chi },raw}$ in an image processing program such as Gimp^TM^ or Adobe Photoshop^TM^ for manual segmentation of the clips.

The resulting $M_{\hat{\chi }}$, created by manually removing any pixel that does not represent a clip or electrode marker from $M_{\hat{\chi },raw}$, is used in combination with $M_{I}$ to extract the pixels that are part of the electrode clips and markers visible on each 16-bit $\chi_{rg}$ image. Any pixel that does not correspond to a clip, is over- or underexposed, or meets the condition in (Equation 3) is not further considered in the following calibration steps. From all other pixel values, a 2D heat map $N_{H}$ with 256 bins for red *r* and green *g* chromacity values each is generated and median-filtered using a 7 by 7 neighborhood.

The red and blue color shades of the electrode markers appear as distinct, Gaussian-shaped peaks $P_{H}$ on $N_{H}$. They (1, 2) are clearly visible as bright spots on the heat map, as shown in Figure 6. A Gaussian mixture model [37,38] is used to extract the individual clusters $C_{H}$ that represent each peak. Each peak is described as a 2D Gaussian distribution, which can be characterized by its center point or centroid and the standard deviations along each direction with respect to this center. By fitting the individual Gaussian models to the heat map N−H, the actual position, orientation, and area covered by each peak can be found. To compute the initial positions of the cluster centroids, the heat map is binarized and labeled. In this process, any 4-connected set of at least 5 bins is considered a peak if all bin counts $n_{H}$ conform to the following condition:(36)$$\begin{array}{c} n_{H}>\left|n_{H}\right|+1.9\sigma \left(n_{H}\right)\;\mathrm{with}\;n_{H}=\left\{n_{H}|n_{H}>0\right\} \end{array}$$

The cluster $C_{H,r}$ with the highest mean $\bar{r_{r}}$ red component is used to compute $\sigma (r_{r})$, $\sigma (g_{r})$, and $\phi_{r}$. The values for $\sigma (r_{b})$, $\sigma (g_{b})$, and $\phi_{b}$ are derived from $C_{H,b}$ for which $1-\bar{r_{b}}-\bar{g_{b}}=max$ holds.
(37)$$\begin{array}{c} \begin{array}{cc} \Sigma_{r}\;U_{r}=\mathrm{eig}\left(\mathrm{cov}\left(C_{H,r}\right)\right)\; & \;\Sigma_{b}\;U_{b}=\mathrm{eig}\left(\mathrm{cov}\left(C_{H,b}\right)\right)\\ \sigma \left(r_{r}\right)=\sqrt{\sigma_{1,r}},\;\sigma \left(g_{r}\right)=\sqrt{\sigma_{2,r}},\;\phi_{r}=\frac{\langle u_{1,r}|r\rangle }{|u_{1,r}|}\; & \;\sigma \left(r_{b}\right)=\sqrt{\sigma_{1,b}},\;\sigma \left(g_{b}\right)=\sqrt{\sigma_{2,b}},\;\phi_{b}=\frac{\langle u_{1,b}|r\rangle }{|u_{1,r}|} \end{array} \end{array}$$
$\sigma_{1,r}$, $\sigma_{2,r}$, $\sigma_{1,b}$, and $\sigma_{2,b}$ represent the first and second eigenvalues Σ of the covariance matrices $\mathrm{cov}(C_{H,r})$, $\mathrm{cov}(C_{H,b})$ of $C_{H,r}$ and $C_{H,b}$, and $u_{1,r}$ and $u_{1,b}$ are the corresponding initial eigenvectors. These values are stored along with the centroids of $C_{H,r}$ and $C_{H}$, which define the mean values $\bar{r_{r}}$, $\bar{g_{r}}$, $\bar{r_{b}}$, and $\bar{g_{b}}$ on disk that are to be used in the extraction step described in Section 2.3.3.

The remaining clusters 3 and 4 are not further considered as they correspond to the color highlights on the clips (3) or are caused by inappropriately chosen parameters affecting the conversion of the raw sensor signals to the RGB color space (4).

### 2.4. Recording Protocol

The technical approach outlined in Section 2.2 and Section 2.3, requires that the patient maintains the same posture throughout the recording. This is only possible if the patient is directly engaged and actively participating in the measurement.

Therefore, prior to the application of the electrodes, the patient is instructed to sit down on a chair. The height of the chair is then adjusted so the patient can comfortably sit upright throughout the recording process. The feet of the patient should rest flat on the floor and the knees should be bent by no more than 90 degrees. If the chair cannot be adjusted in height, an alternative solution is to stack multiple chairs to increase the patient’s comfort and encourage them to straighten their back. To ensure optimal recordings without any obstacles, the chair should not have armrests or a backrest and be placed at least 1 meter from any furniture or other objects that can cause shadows. This ensures that the FOV of the 3D DS camera can be optimally used and the operator is able to capture a surface at least every 20 degrees.

After the electrodes have been attached to the torso, the patient is instructed to place the hands on the thighs. The fingers should point inward and the thumbs should point straight toward the hips. The optimal position of the hands is a thumb length before the hips. While the electrode positions are recorded, the patient is instructed to maintain a straight and upright back. Most patients are able to easily maintain this position by slightly straightening their elbows (about 120 degrees between the upper and lower arm). This helps them to move their chest and shoulders into a position that is as upright as possible. This has the effect that the patient is forced into an isometric posture, which can easily be maintained while the electrode positions are recorded. In addition, this position facilitates the recording of electrodes placed under the left axle, for example, the Wilson electrodes $V_{5}$ and $V_{6}$.

## 3. Results

In the following section, the results are presented.

The narrow vertical field of view of the color sensor is one of the main reasons why the 3D images of the torso are recorded in portrait mode. In a typical clinical setting, where space is limited, it is likely that the patient is seated close to furniture or walls. For proper recording of the 3D images, a space of at least 2.5 m by 2.5 m is required. This includes a standard chair without armrests or a backrest, with a diameter of 50 cm, that can provide at least 1 m of space on all four sides of the patient for the operator to move around while recording the images. The remaining space between the patient, the operator, and any surrounding furniture or walls should be 50 cm or less. Both sensors of the camera must be able to properly capture the dorsal part of the patient’s torso at distances between 20 cm and 50 cm. This can only be achieved by cameras with FOV angles conforming to (Equation 1) such as the Intel Realsense cameras, which have wide viewing angles of ≈70 degrees for both the depth and color sensors when used in portrait recording mode. This is especially important for capturing the dorsal views of the torso.

The color sensor has a viewing ratio of 16:9 between the horizontal and vertical FOVs. This results in a vertical viewing angle of about 40 degrees, which is a lot smaller than the ≈60 degrees of the depth sensor. This can lead to a situation where, for example, around ≈60 columns on the top and bottom of the depth image lack texture information. However, this is acceptable given that consecutive 3D images are recorded in portrait mode with an overlap of about two-thirds, ensuring that the texture images sufficiently overlap.

Thanks to the vertical nature of the patient’s torso, in portrait mode, it is easy to keep the patient centered in the image while moving the camera to the next recording position. As the patient’s torso covers most of the image space, only very few objects and obstacles located behind the patient are captured by the cameras, which can easily be removed before storing the 3D surface images.

Scanning always starts with the right frontal view of the torso and ends at the right dorsal side. If possible, the right lateral side of the torso can be recorded. This is not essential for extracting the electrode positions and can be omitted in standard recording procedures. It is recommended to explicitly record the right lateral torso surface when there is sufficient space to the right of the patient.

The preview image of the torso, shown in the main area (1) of the user interface shown in Figure 7, is split into a 3-by-3 grid. The center segment of this grid is used as the focus area, representing the central part of the patient’s torso. The contours of the largest object containing the focus segment are highlighted in orange. As the camera points at the patient’s torso, the contours highlight the boundaries of the patient’s torso. The recording of a torso surface segment is initiated by pressing the trigger of the camera. The color of the contour line switches to green and the live preview freezes to indicate that the captured depth and color image have been processed and the 3D surface has been generated and stored. Once the underlying point cloud has been triangulated, occluded and degenerated triangles, as well as detached surface patches, are removed. Then, the contour is updated to mark the parts that will be stored on disk. After the 3D surface information, corresponding texture image, and meta information have been stored, the live preview is started again and the color of the contour reverts to orange. The live preview is updated at a maximum rate of 10 FPS. With the Python-based prototype, update rates between ≈4 FPS and ≈7 FPS can be realistically achieved.

The main preview area (panel 1 in Figure 7) has the same shape as the depth image. For the parts on the left and right sides that are not captured by the RGB image, the edges identified on the depth image are displayed instead. The outline of the patient’s torso does not extend beyond the edges of the RGB image. In panel 2 of the preview screen (Figure 7), several recording and camera parameters, such as the frame rate in FPS, exposure time τ in ms, etc., are shown, along with the intermediate parameters computed for automatic-exposure control and color correction. In panel 3, the full set of edges identified on the current depth image is displayed. The two vertical lines delineate the area of the depth image that is covered by the color image.

The prototype for real-time recording of the 3D torso surface patches, as well as for postprocessing and calibration, was implemented in Python version 3 using a recent version of NumPy and SciPy [39]. The librealsens version 2 library [40] was used to control the acquisition, convert the depth values into a point cloud, and compute the corresponding texture uvs map for the RGB image. The OpenCV library [41] was used to generate the preview display, and the generation and cleaning of the 3D meshes were accomplished using the open3D library [27]. The most computationally demanding components, the depth-edge detection (Section 2.2.3), automatic white balancing (Section 2.2.1), and patient-locked auto-exposure control (Section 2.2.2), were converted into Python-C modules using Cython [42].

In total, five male subjects between 38 and 70 years of age participated in the present study. Each subject was seated on a chair or examination bed, depending on the available space. After applying the ECG electrodes to the chest and back, the subjects were instructed to maintain the posture described in Section 2.4. The measurement of the torso surface and the recording of a 30 -min long ECG with 67 channels took about 30 min to 45 min. After each measurement, the data were analyzed and the prototype improved accordingly.

The data set recorded from the first subject turned out to be quite limited and, therefore, is not included in the presented results, as it was affected by the automatic white balancing and exposure control of the color sensor, which could not cope well with the diverse and complex lighting conditions. Further, the 3D points recorded by the depth sensor were directly transformed to match the color image captured by the color sensor. This posed several challenges related to occluded surface parts causing undesirable distortions and the introduction of noncausal surfaces. Starting with the data for the second subject, the direct mapping was replaced with the texture mapping approach, which yielded better results and allowed for the implementation of the algorithms for occlusion management and the removal of noncausal triangles, as described in Section 2.2.4.

For each patient, 12 to 15 views were recorded. Each of the views contained a 3D surface described by ≈170,000 vertices and ≈300,000 triangles. As shown in Table 1, between 7 and 21 iterations of the symmetric ICP algorithm were necessary to align the surfaces. The maximum correspondence distance between the points of the surface pairs was reduced in every iteration step, starting from 7 cm–12 cm and reaching 0.7 mm–1.2 mm. More iterations were necessary to align the surfaces joining the frontal and dorsal views on the left side of the torso. In cases where the available space around the subject was insufficient, the number of iterations required to align the surfaces was increased. In the most challenging scenario, the proper alignment of the surfaces was not possible at all. This situation was encountered in the data set recorded from subject 5, where part of the torso surface on the left side was obscured by the backrest of the chair. Among other challenges, this required an increased number of 21 iterations to align the leftmost frontal and dorsal views.

Across all subjects, a final root mean square error between consecutive surfaces of 0.7 mm was achieved. Using the proposed approach, 12 to 15 surfaces per patient were registered within 13 min. As shown in Table 2, the extraction of the electrode marker points and the computation and labeling of the electrode positions were completed after another ≈8 min.

The recording sessions were part of a larger clinical pilot study investigating the prognostic value of index arrhythmias with respect to the outcome of pulmonary vein ablation, for which the participants provided informed consent. Apart from the 3D camera and ECG recordings, this study was based on clinical data recorded during the patient’s clinical treatment. Therefore, CT recordings and other independent means of recording the electrode positions relative to the torso were included. To assess the accuracy of electrode localization, the electrode positions were backprojected onto the individual views of the torso and marked on the corresponding color images. Examples are shown in Figure 3, Figure 4b,c and Figure 8.

The annotated RGB images were presented to an expert who used the cross-hair tool shown in Figure 8b to manually adjust the position of each marker. In order to facilitate this task, two markers were used: the green marker indicates the backprojected position of the marker and the red marker corresponds to the manually adjusted position. All positions were checked during this process and if necessary, they were moved to better reflect the perceived center positions on each view. When finished, all positions were reprojected onto a 3D space.

For the set of corrected positions of each electrode, the mean point, as well as the mean distance and standard deviation to this mean point, were computed. The resulting values are shown in Table 3, along with the mean and standard deviations of the computed electrode positions with respect to the manually determined mean. Both sets of results were influenced by the accuracy of the registration process and the fact that no unique solution exists for the backprojection of the electrode positions onto the individual views. In addition, the mean and standard deviation of the registration errors and the error between the mean and standard deviation of the distances between the individual projections and their mean point are listed.

The corrected electrode positions deviated, on average, by 2.3mm±1.4mm from the mean point, and the computed electrode positions deviated from the mean point by [2.0mm±1.5mm]. This is in accordance with the limitations posed by the backprojection, where the reprojected points deviated from the computed position by 0.9mm±1.4mm, and the ICP registration resulted in an average deviation between corresponding points of 0.6mm±0.2mm. Given the amount of data to be processed per subject, the overall time of 22 min. required to extract and align the electrode positions is quite impressive, considering that only the computations of the asymmetric ICP and the HDBSCAN algorithms are implemented as part of the native open3D library and as Cython scripts, respectively. The rest of the implementation was carried out in Python using NumPy arrays only. In contrast, the expert required between 30 min. and 45 min. to point and place the electrode markers on the 14 views of a single data set.

## 4. Discussion

The results are promising given the fact that the torso is a far less rigid structure compared to the skull. Further, the limited space conditions and adverse environmental conditions typically found in clinical settings, e.g., outpatient and local practitioner clinics, are quite challenging. This is evident in the results shown in Table 3 for subjects 4 and 5. In both cases, nearby obstacles such as backrests or furniture limited access to the patient’s left side, resulting in increased positional variations of 2.2 mm±1.5 mm and 2.4 mm±1.8 mm in relation to the mean of the manually defined electrode positions. This is compared to 1.7 mm±1.4 mm and 1.6 mm±1.5 mm for subjects 2 and 3, respectively.

These values are still in the range reported for recently proposed approaches for localizing electrodes mounted on the human body. As shown in Table 4, few studies exist that evaluate the use of 3D DS cameras [19,20] and photogrammetry methods [18] for localizing ECG electrodes on the torso. The achieved results varied between 1.16 mm and 11.8 mm, depending on the metrics and positional references used. The authors of [20] used the Hausdorff metric to compare the positions obtained from a Microsoft Kinect 3D DS camera to positions found on MRI or CT scans. On average, they achieved a positional error of 11.8 mm, which is an order of magnitude larger than the error between 1.16 mm and 2.5 mm achieved by Schulze et al. [18], Alioui et al. [19] and the present study, all of which used the Euclidean metric instead.

The majority of studies proposed methods for the localization of EEG sensors mounted on the scalp. Apart from Homölle and Oostenveld [8], the achieved average positional errors ranged from 1.5 mm [12] to 3.26 mm [14] using various reference measurements, including the mean of manually placed marks [12,14] and positional references generated using a magnetic digitizer [8,13,16] such as the Polhemus Fastrak. Comparing the positional error of 9.4 mm achieved by Homölle and Oostenveld [8] to all other results, it can be assumed that this was mainly caused by unavoidable inaccuracies when taking the magnetic digitizer measurements.

Considering that the positions of ECG electrodes mounted on the torso are directly affected by any movements, the positional error of 2.0 mm achieved in the present study is a clear indication that the active engagement and participation of the patient in the measurement is essential. The instructions on how the patient can easily maintain a posture that facilitates the recording of the electrode positions have a huge impact on the outcome of the measurements. If the instructions are not clearly defined by the measurement protocol, or not properly understood or followed by the patient, the positional error will increase. For example, subject 4 (see Table 3) changed the position of his arms during the measurement twice. This immediately resulted in an increased positional error of 2.2 mm ±1.5 mm.

In addition to the limited space, the lighting conditions encountered in the clinical environment, as well as tight schedules, have a direct impact on the average positional error. Varying lighting conditions, including multiple light sources with differing light temperatures, on the other hand, can have a negative impact on photogrammetric approaches and 3D DS camera-based measurements of the torso surface and the electrode positions thereon. Algorithms for automatic white balancing and exposure control have been adopted to improve color constancy across multiple 3D views of the torso and maintain a constant exposure of the torso independent of the viewing direction and angle. In combination with the developed calibration method, this resulted in increased accuracy in identifying those pixels representing the color markers.

Time, in particular, is a very limited resource, which largely limits the routine use of magnetic digitizers within clinical environments. For precise positional measurement, the exact placement of the magnetic probe on each electrode and manual triggering of the measurement are required. An experienced user requires about 15 min. to accomplish this task. Any attempt to reduce this time can only be achieved by the less accurate placement of the probe on each electrode, which can result in increased positional errors of 7.8 mm and higher, as encountered by Clausner et al. [43].

In general, keeping the required human interactions and number of related errors as low as possible is one key goal for establishing NICE-based tools and procedures in clinical environments. The time required to localize the electrode positions on the human torso, as well as the amount of ionizing radiation the patient is exposed to, are key factors that can either prevent or facilitate a successful uptake. Alternative approaches currently used to obtain the electrode positions include manually placing markers on CT and MRI scans [9,12,19]. and automatically segmenting and pointing a magnetic digitizer probe to each individual electrode [8,13,16]. These approaches require a significant amount of time (about 45 min.) to point to each electrode, which is more than the 15 min. required for magnetic probe-based measurements. The mentioned approaches suffer from an additional bias related to the individual human perception of the electrode and marker shapes, as well as inaccuracies in the way the pointing probes are placed onto the electrode.

In contrast, the proposed 3D DS camera-based approach is not affected by these kinds of errors. When implemented on a tablet computer, the presented approach will enable clinicians to acquire the electrode positions and torso surfaces within 10 min. Therefore, average positional errors of less than 2.5 mm will be feasible even under limited spatial conditions and tight schedules.

Some aspects essential for the successful clinical uptake of the presented approach still have to be addressed. On all color sensors, the raw signals recorded for red, green, and blue channels have to be converted into the RGB color space before they can be used. If the required parameters are not properly calibrated, the resulting images may show a bluish hue that can not be corrected by any white-balancing algorithm. This was the case for subject 5 shown in Figure 3, and caused the additional peak (4) in the calibration heat map shown in Figure 6. During the preparation of future studies, it is necessary to establish an appropriate procedure for verifying and optimizing the settings for these parameters before the first measurement and at regular intervals.

Each 3D DS camera data set also provides a point cloud representation of the torso surface. This is used in current studies to build electroanatomical models for electrocardiographic noninvasive imaging methods from clinical cardiac CT slices only. Further applications of the proposed approach are currently being investigated for enhanced electrical impedance tomography.

## 5. Conclusions

In the presented work, a complete 3D DS camera-based system was developed for localizing 67 ECG electrodes identified by color markers. Issues such as varying lighting conditions, including multiple light sources with different light temperatures, and the alignment of individual 3D views were addressed. The implemented recording protocol provides precise rules on how to seat the patient and includes well-defined instructions for the patient to easily maintain a specific isometric posture while all views are recorded. The resulting active engagement and participation of the patient in the measurement helped to minimize positional errors caused by the patient moving during the measurement. In combination with the symmetric ICP algorithm implemented, average positional errors of 2.3 mm or less could be achieved for each measurement.

The implemented prototype system localizes the electrodes on the torso with minimal human interaction. It can handle diverse lighting conditions and operate in narrow spaces, as encountered in clinical settings such as outpatients of local practitioner clinics.

## Figures and Tables

**Figure 1 sensors-23-05552-f001:**
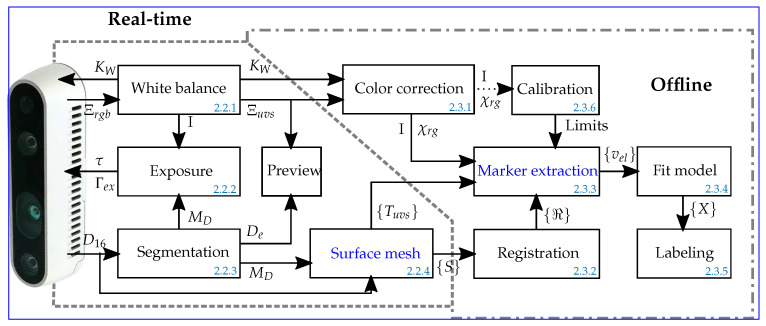
Schematic presentation of the overall approach used to extract ECG electrode positions from 3D depth-sensing camera data. The first step involves methods to control the camera’s white balance and exposure settings and generate textured 3D surface meshes from the recorded depth data. During the offline processing step, these surfaces are aligned to extract the electrode positions within clusters of marker vertices found using texture images in the RGB chromacity color space and 3D surfaces.

**Figure 2 sensors-23-05552-f002:**
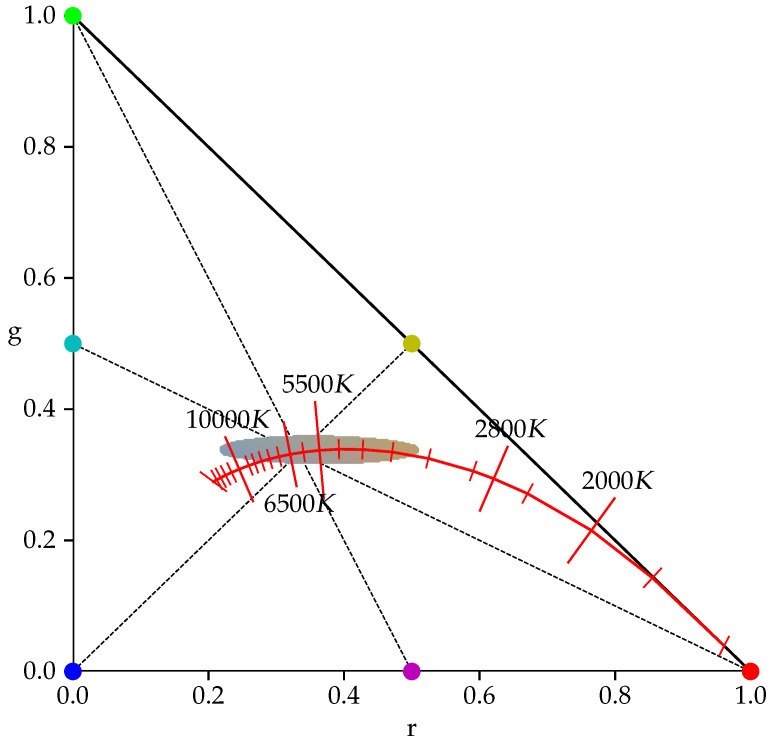
The elliptic area in the RGB chromacity space corresponds to the pixels encoding shades of gray [23]. The red *r* and green *g* chromacity values of the natural illumination color gamut of 5500 K define a point that is shifted slightly off the mean RGB chromacity toward yellowish colors. Standard 3D DS cameras designed for indoor use such as the Intel Realsense^TM^ typically allow adjusting the gains for the red and blue channels to illumination color gamuts between, for example, 2800 K and 6500 K, as indicated on the color gamut curve. However, in real clinical settings, gamuts from 2000 K up to 10,000 K can be expected, depending on the number of light sources and shades cast by objects and people.

**Figure 3 sensors-23-05552-f003:**
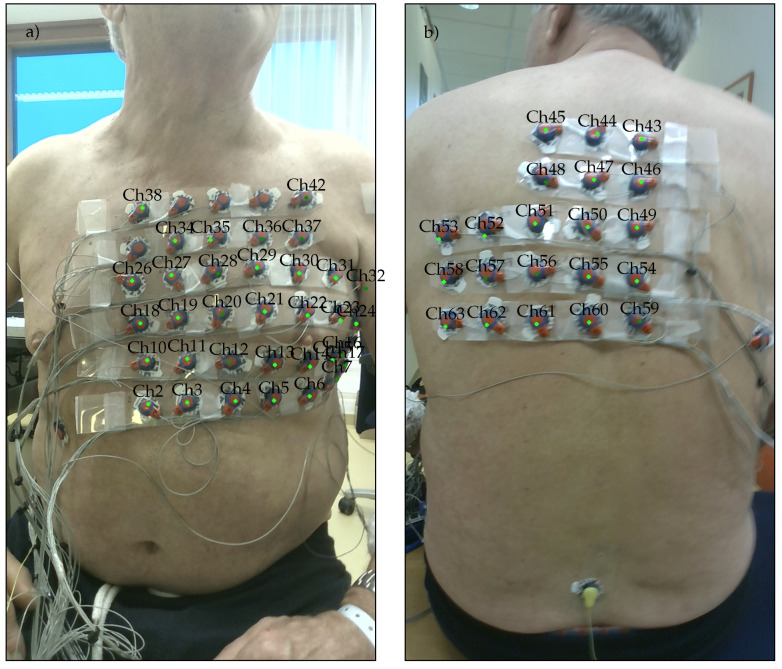
The histogram-based auto-exposure algorithm considers only the pixels that most likely correspond to the patient and ignores any others. This ensures that the brightness of the patient’s skin remains as constant as possible, regardless of whether the camera points toward a window (**a**) or the darkest corner of the room (**b**). Each visible electrode is labeled with the corresponding channel number and the ‘+’ markers indicate the projected location of the computed electrode position.

**Figure 4 sensors-23-05552-f004:**
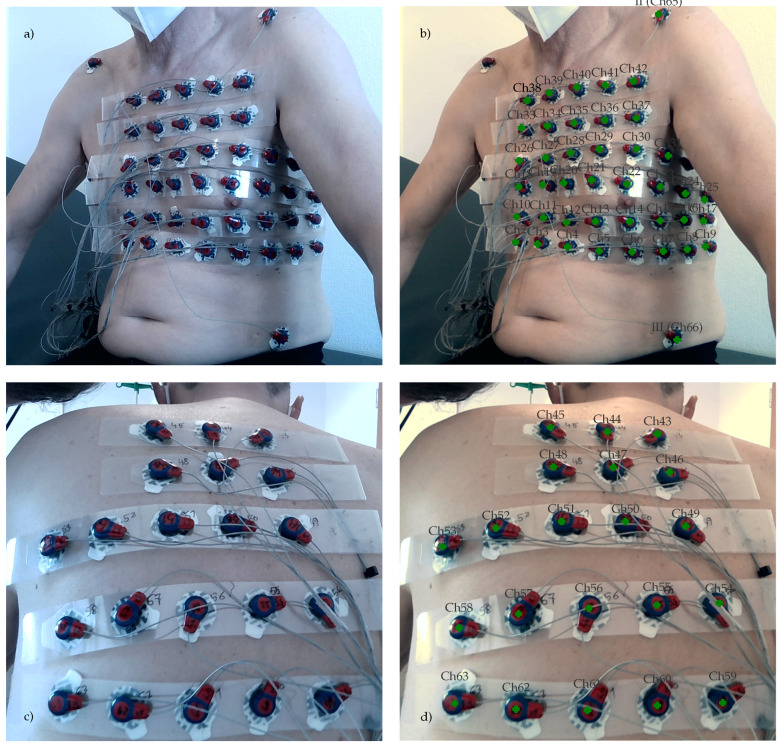
Examples for texture images recorded from the front (**a**) and back (**c**) of the torso, and the corresponding color-corrected versions (**b**,**d**). In (**b**,**d**), each visible electrode is labeled with the corresponding channel number. The ‘+’ markers indicate the projected location of the computed electrode position.

**Figure 5 sensors-23-05552-f005:**
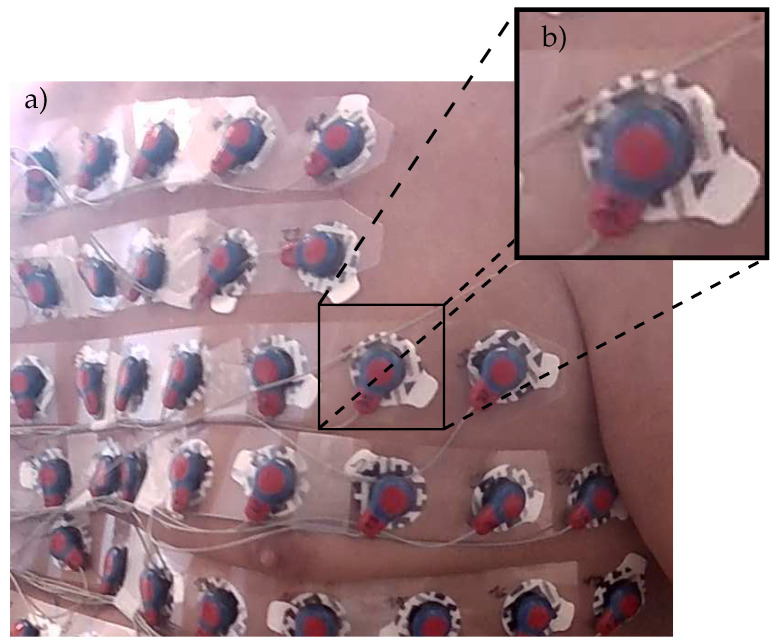
The electrodes mounted on the patient’s torso (**a**) are attached to the red electrode clips. The blue boundary of each clip head (**b**) forms a circular marker with the red electrode clip. Based on the red and blue colors, each marker can be recognized from the recorded texture images along with 3D surface information.

**Figure 6 sensors-23-05552-f006:**
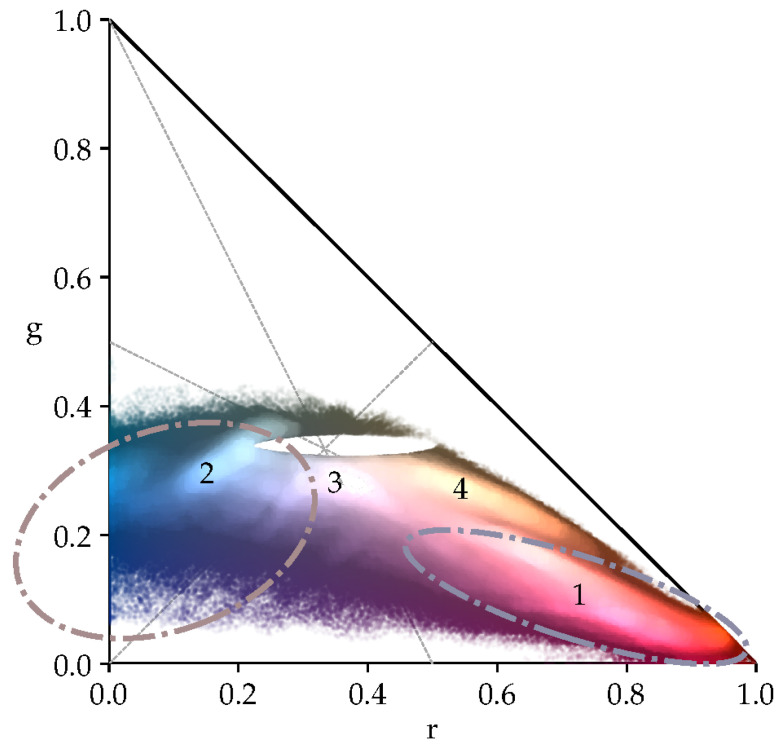
Chromacity heat map of the pixels representing the electrode markers created from the texture images of three patients. The brighter the color, the higher the pixel count for the corresponding point in the RGB chromacity space, represented by its red *r* and green *g* chromacity values. For better readability, the RGB chromacity values are displayed in RGB gamma-compressed form. The Gaussian peaks (dash doted ellipses) representing the red (1) and blue (2) pixels of the electrode markers are clearly visible. They can easily be distinguished from the peak (3) representing the color highlights and reflections. Peak (4) is caused by inappropriately chosen values for the parameters required to convert raw color sensor data to the RGB color space.

**Figure 7 sensors-23-05552-f007:**
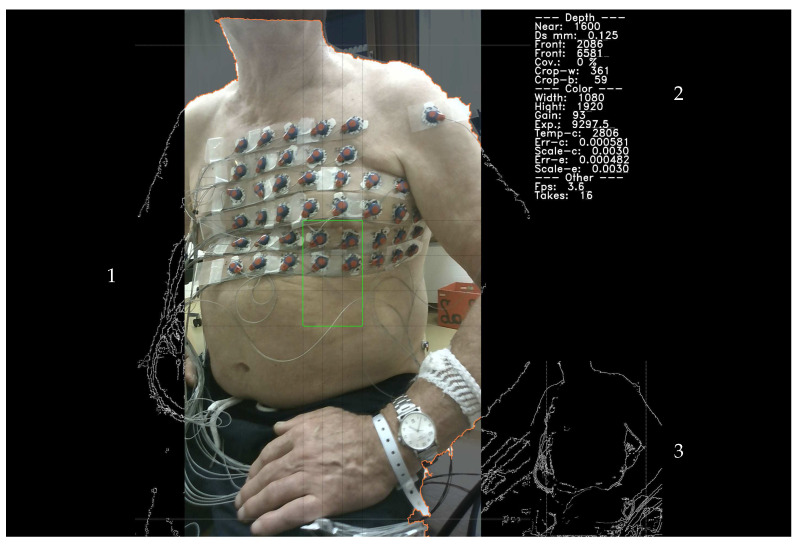
The preview screen is divided into three panels. The main panel (1) displays the image recorded by the color sensor of the 3D DS camera. The parts of the 3D image for which no color information could be captured are replaced with the edges extracted from the depth image shown in panel 3. The current values of the color temperature, exposure time, frames per second, and other process parameters are displayed in panel 2. The two vertical lines indicate the area where the views of both cameras overlap.

**Figure 8 sensors-23-05552-f008:**
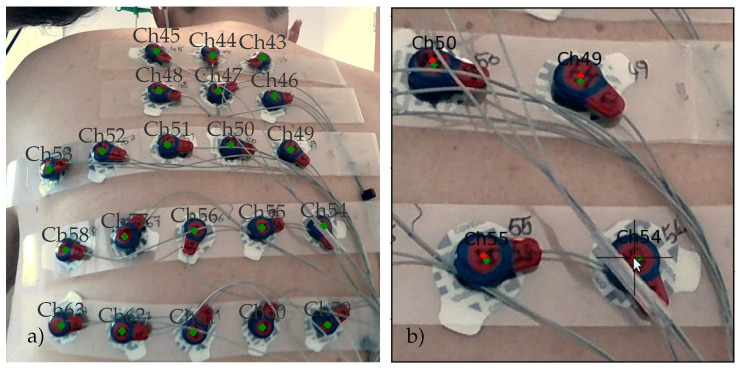
Manual evaluation of the proposed approach for locating the electrode positions on a patient’s torso. The electrode positions are backprojected onto each recorded torso surface segment (**a**). An electrode position can be moved by clicking on the corresponding green cross-shaped graphical marker displayed on the texture image. Its new position is selected by pointing and clicking on it (**b**). In case the position pointed to is not backed by a valid surface triangle, the new point (red cross) is moved to the closest possible position. By right-clicking on an electrode marker, it can be disabled and/or enabled on the presented view. Any disabled markers are not considered suitable for further evaluation.

**Table 1 sensors-23-05552-t001:** The symmetric ICP alignment metrics obtained from the data sets of four out of five subjects participating in the study. For an average of 14 angular views, the computation of the pairwise transformation matrix was repeated between 7 and 21 times, with an average repetition rate of 9.7 repeats per surface pair. The average initial distance between corresponding points was 7 cm, the average root mean square error was 0.7 mm, and the average final correspondence distance was 1 mm.

Subj.	# Views	# $\frac{\mathrm{repeats}}{\mathrm{pair}}$			Correspondence Distance (mm)						Final Rmse (mm)		
					Initial			Final					
		Min.	Mean	Max.	Min.	Mean	Max.	Min.	Mean	Max.	Min.	Mean	Max.
2	12	9	10.1	12	37.4	71.6	124.9	0.7	0.9	1.2	0.5	0.6	0.8
3	14	7	9.4	17	51.4	76.1	105.0	0.9	1.0	1.1	0.6	0.7	0.7
4	14	8	9.4	17	42.9	61.2	105.2	0.9	1.0	1.1	0.6	0.7	0.8
5	15	8	9.9	21	50.0	71.3	117.2	0.9	1.0	1.1	0.6	0.7	0.7
Mean	14	7	9.7	21	37.4	70.0	124.9	0.7	1.0	1.2	0.5	0.7	0.8

**Table 2 sensors-23-05552-t002:** Performance measures obtained from four out of five subjects participating in the study. The positions mounted on a patient’s torso were extracted from 14 recorded angular views, on average, within ≈21.8 min. The symmetric ICP-based pairwise alignment of the corresponding surfaces of about 139,527 vertices and 254,351 triangles took approximately≈23 of this time.

Subj.	# Views	# Vertices			# Triangles			Time (min.)		
		Min.	Mean	Max.	Min.	Mean	Max.	ICP	Pos.	Total
2	12	63,204	101,446	132,813	112,547	178,654	235,112	6.58	5.75	12.51
3	14	131,090	198,732	232,417	252,514	380,923	448,766	17.75	9.94	28.28
4	14	155,025	189,075	223,063	292,232	357,555	426,261	14.62	7.78	22.94
5	15	172,548	203,327	241,228	322,431	384,722	458,542	15.24	7.79	23.48
Mean	14	63,204	176,301	241,228	112,547	331,879	458,542	13.55	7.82	21.80

**Table 3 sensors-23-05552-t003:** The electrode positions computed using the proposed approach and defined by manually marking the clip center on each view deviated from each other, on average, by ≈1.9 mm ± 1.5 mm. In addition, the distance between the computed electrode position and the mean point obtained by the projection of each electrode onto each individual view, as well as the average position found by manually marking the center of the clip on each view, are provided. Along with the variation resulting from the ICP-based surface alignment, both values allow for the assessment of how well the proposed approach can approximate the true positions of the electrodes.

Subj.	Manual (mm)		$\bar{\mathrm{Manual}}-\mathrm{marker}$ (mm)		Mapping (mm)		Registration (mm)	
	Mean	std.	Mean	std.	Mean	std.	Mean	std.
2	1.4	0.9	1.7	1.4	0.6	0.9	0.6	0.2
3	2.1	1.4	1.6	1.5	0.9	1.4	0.6	0.2
4	2.9	1.8	2.2	1.5	1.1	1.8	0.7	0.2
5	2.7	1.6	2.4	1.8	0.9	1.6	0.6	0.2
Mean	2.3	1.4	2.0	1.5	0.9	1.4	0.6	0.2

**Table 4 sensors-23-05552-t004:** Comparison of the proposed approach for localizing ECG electrodes on the torso using a 3D DS camera with recent developments. In contrast to the large number of publications addressing the localization of EEG electrodes, only a few could be found using 3D DS cameras. The results obtained in the present study are within the ranges found by other studies.

Source	Sensor/Method	Multi-View	Ref.	# El.	μ	σ	max.
		Registration			(mm)	(mm)	(mm)
ECG							
Presented	3D DS	Symmetric ICP	manual mean	67	2.0	1.5	–
			reprojection		0.9	1.4	–
	manual marking		manual mean		2.3	1.4	–
Perez E. (2018) [20]	3D DS	Kinect Software	MRI/CT	128	11.8	–	64.8
Alioui S. (2017) [19]	3D DS	only one view	N/A	3	2.5	–	–
Schulze W. (2014) [18]	Photogrammetry	Least Squares	marker plate	80	1.16	0.97	–
EEG							
Chen S. (2019) [14]	3D DS	Least Squares	multiple repeats	30	3.26	1.05	–
Homölle S. (2019) [8]	3D DS	Scanner Software	Magnetic digitizer	61	9.4	–	10.9
Cline C. C. (2018) [13]	IR-Scanner	Scanner Software	Magnetic digitizer/	128	1.73	0.37	–
	Magnetic digitizer		IR-Scanner		2.98	0.89	–
	VR-Digitizer				3.74	0.71	–
Clausner T. (2017) [43]	Photogrammetry	Photogrammetry	Face Scan	68	1.30	0.6	–
	Magnetic digitizer	software			7.80	2.1	–
Butler R. 2017 [9]	MRI	–	manual mean	63	0.5	–	–
DalalS. S. (2014) [12]	Magnetic digitizer	–	FaceScan +	68	6.8	–	13.3
	Flying Triangulation		manual mean		1.5	–	2.9
Kössler L. (2010) [16]	Laser scanner	Scanner software	Magnetic digitizer	68	1.83	1.16	–

## Data Availability

The data and software are made available in the data and software repository of Johannes Kepler university. Until public access to these reporsitories is available in its full extent data developed data and software are provided on request by the institute for Biomedical Mechatronics at Johannes Kepler University, mmt@jku.at.

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
