# Peer review of "ECG Electrode Localization: 3D DS Camera System for Use in Diverse Clinical Environments"

_sensors, 2023, doi:10.3390/s23125552_

Round 1
Reviewer 1 Report
The present study, entitled "ECG electrode localization: a 3D-DS camera system for use in diverse clinical environments," is valuable work. The issue of the manuscript is about presenting a 3D-depth sensing camera-based approach for recording electrode positions that have been established. The developed prototype can handle diverse lighting conditions when space is limited, which is typically encountered within clinical settings like outpatient clinics. It is an attractive topic and can be useful in recording ECG. But by reading the manuscript, questions exist that need further explanation. For this reason, I am considering major revisions for this manuscript and stating the necessary questions and corrections as follows that should be considered:
1. The text of the manuscript needs corrections that can be reviewed by a native English speaker.
2. In the abstract, the gap is not well addressed, and it needs to be rewritten.
3. Why does it appear as an unsharp shadow in a color image for any object closer than dnear? Expressing this issue requires more explanations.
4. In lines 123 and 124 of the manuscript, the authors stated that the depth sensor of this camera could take pictures of objects that are between 20 and 150 cm away from the camera. On what basis is this distance obtained? And how do the changes in this distance affect the measurement of the depth sensor?
5. When using an infrared sensor, is its wavelength investigated?
6. The methods part of the manuscript has been written very long, so it needs to be reduced to make the method clear so that it is not boring for the reader. Also, the article review section should be separated from the method section.
7. In the ICP registration, is the selection of initial points before registration also examined in the results?
8. There is a need for more discussion in the text of the manuscript about comparing the results obtained with similar works, according to Table 4.
Author Response
Pleas see the attachment.

Reviewer 2 Report
The submitted article adeptly addresses the utilization of a 3D camera for surveying the position of electrodes in patients during medical examinations. The article covers various criteria for selecting the appropriate camera and discusses the processing of data and images.
The article lays a solid foundation and is generally well-written. However, there are some topics that could benefit from improvement to enhance the overall quality.
In the abstract, line 7 states, "Most of the models include additional signals like electro encephalo gram (EEG) or electro cardio gram (ECG) and thus require that the corresponding sensor positions, typically between 12 and a few hundred, are exactly known. The larger the positional error the lower is the diagnostic value of the resulting model. Actually, position accuracy is crucial for interpreting other results and exams, not ECG or EEG. Although this is clarified later in the Introduction, the abstract text is unclear on this point.
Figure 1 contains several elements that are not described in the text and do not contribute to understanding the materials and methods discussed. It would be beneficial to revise the figure to align more closely with the text and enhance overall coherence.
Please, consider using electroencephalogram and electrocardiogram instead of "electro encephalo gram (EEG) or electro cardio gram (ECG)"
